# Differential cell survival outcomes in response to diverse amino acid stress

Marion Russier[1] , Alessandra Fiore[1], Ana Bici[1] , Annette Groß[1] , Maria Tanzer[2], Assa Yeroslaviz[3] , Matthias Mann[2] , Peter J Murray[1]

Amino acid (AA) detection is fundamental for cellular function, balancing translation demands, biochemical pathways, and signaling networks. Although the GCN2 and mTORC1 pathways are known to regulate AA sensing, the global cellular response to AA deprivation remains poorly understood, particularly in non-transformed cells, which may exhibit distinct adaptive strategies compared with cancer cells. Here, we employed murine pluripotent embryonic stem (ES) cells as a model system to dissect responses to AA stress. Using multi-omics analyses over an extended time course, we examined the effects of arginine (Arg) and leucine (Leu) deprivation. We uncovered a broad array of proteomic, phosphoproteomic, transcriptomic, and metabolomic adaptations, including an increase in lysosome production, all occurring without lethality. We found that Arg or Leu starvation induces reversible cell cycle exit, promoting a quiescent state that enhances resistance to cytotoxic stressors. In contrast, cysteine (Cys) and threonine (Thr) deprivation led to cell death via distinct pathways: ferroptosis for Cys starvation, whereas Thr deprivation triggered a previously uncharacterized form of cell death, which could be entirely suppressed by methionine (Met) co-starvation, and mTOR or translational inhibition. These findings suggest that ES cells implement specialized survival strategies in response to different AA limitations, highlighting their ability to reprogram cellular biochemistry under nutrient stress.

## Introduction

Cells continuously monitor amino acid (AA) availability to regulate metabolism, translation, and growth (Morris & Rogers, 1978; Zhu & Thompson, 2019; Gu et al, 2022). AAs are sourced from different "pools," including external uptake via solute carrier (SLC) transporters, intracellular reservoirs such as lysosomes, biosynthetic pathways, and protein degradation. In transformed cells, increased proliferation alters AA demands, leading to adaptive strategies to obtain AA, such as enhanced autophagy and ribophagy, increased solute transporter expression, uptake of abundant serum proteins (Commisso et al, 2013; Kamphorst et al, 2015; Palm et al, 2015; Nofal et al, 2017, 2022; Armenta et al, 2022), and catabolism of local matrix proteins or oligopeptides (Nazemi et al, 2024; Guzelsoy et al, 2025) or secretion of specific AAs (Sousa et al, 2016). Key outstanding questions in AA metabolism include how cells detect individual AAs, assess their proportional balance, and integrate this information to regulate cellular pathways.

Two major AA-sensing pathways, GCN2 and mTORC1, mediate adaptive responses to AA availability (Saxton et al, 2016a, 2016b; Chantranupong et al, 2016; Wolfson et al, 2016; Wolfson & Sabatini, 2017; Costa-Mattioli & Walter, 2020). GCN2 activation, triggered by uncharged tRNAs at stalled ribosomes, initiates the integrated stress response (ISR), reducing translation through eIF2α phosphorylation (Ishimura et al, 2016; Inglis et al, 2019; Masson, 2019; Misra et al, 2021, 2024). Conversely, mTORC1, activated by AA-binding proteins such as Sestrin-2 (Leu sensor) and Castor-1 (Arg sensor), promotes anabolic metabolism, translation, and ribosome biogenesis (Saxton et al, 2016a, 2016b; Chantranupong et al, 2016; Wolfson et al, 2016; Wolfson & Sabatini, 2017; Lee et al, 2018; Kim & Guan, 2019; Kim et al, 2020; Chen et al, 2021; Cangelosi et al, 2022). These pathways operate antagonistically, with GCN2 sensing AA scarcity and mTORC1 detecting AA abundance. Despite advances in understanding these pathways, major gaps remain in elucidating how mammalian cells collectively detect and prioritize AAs under physiological conditions (Carroll et al, 2016; Cui et al, 2023; Fernandes et al, 2024).

AA sensing and metabolic adaptation are particularly relevant in cancer, where specific AAs become metabolic dependencies. Moreover, AA limitation can drive cells into a quiescent state, rendering them vulnerable to metabolic perturbations (Livneh et al, 2023, 2024; Pu et al, 2023). However, AA sensing is highly

---

[1]Immunoregulation Research Group, Max Planck Institute of Biochemistry, Martinsried, Germany   [2]Department of Proteomics and Signal Transduction, Max Planck Institute of Biochemistry, Martinsried, Germany   [3]Computational Systems Biology Research Group, Max Planck Institute of Biochemistry, Martinsried, Germany

Correspondence: murray@biochem.mpg.de
Marion Russier's present address is Genmab, Utrecht, Netherlands
Alessandra Fiore's present address is Department of Neurosciences, Biomedicine and Movement Sciences, Section of Biochemistry, University of Verona, Verona, Italy
Maria Tanzer's present address is Walter and Eliza Hall Institute, Melbourne, Australia

context-dependent, with different cell types exhibiting unique responses (Shorthouse et al, 2022). Without a defined "baseline" model, interpreting cell-specific metabolic adaptations remains challenging. Here, we use murine embryonic stem (ES) cells as a non-transformed, rapidly proliferating model system (~8- to 10-h division time) to investigate fundamental responses to AA limitation. As ES cells are pluripotent and nonmalignant, they provide an ideal system to elucidate core mechanisms underlying cellular adaptation to AA stress, independent of oncogenic transformation.

Our study aimed to gain an understanding of how non-transformed cells sense and respond to AA deprivation in a global way by leveraging high-resolution proteomic, transcriptomic, and metabolomic approaches. By systematically comparing ES cell responses to deprivation of different AAs, we found both common and distinct mechanisms that govern cellular adaptation to AA stress, including cell survival pathways.

# Results

### Global remodeling of cell state during long-term AA starvation

To gain insights into how ES cells negotiate AA starvation, we employed a global, time-resolved multiparametric approach, using deprivation of the AAs Arg or Leu, which are critical for ES cell proliferation (Correia et al, 2022; Todorova et al, 2024) and both of which activate mTORC1 (Wolfson & Sabatini, 2017). Within hours of starvation, ES cells exited the cell cycle and showed a marked reduction in overall biosynthesis (Fig 1A and B). Importantly, ES cells tolerated Arg or Leu deprivation stress reversibly; even after 48 h of Arg or Leu deprivation, starved ES cells remained viable consistent with other findings (Correia et al, 2022) (Fig S1A). Readdition of either Arg or Leu triggered immediate reentry into the cell cycle as measured by live-cell imaging (Fig 1C). Using quantitative mass spectrometry over 1, 3, and 10 h of starvation revealed that, aside from Arg or Leu, the intracellular levels of other AAs remained largely stable—with the exception of proline, which increased in both conditions (Fig 1D). The reasons behind this increase in proline remain unclear but may point to a yet to be understood role in AA stress adaptation (Liang et al, 2013).

To gain a global view of how ES cells adapted their proteomes to AA stress, we deprived ES cells of Arg or Leu for 1, 3, or 10 h and profiled protein and phosphoprotein abundance in parallel with RNAseq analysis at 3 h, when cells ceased proliferation (Fig 1E). Principal component analysis revealed a broad reorganization of the proteome, phosphoproteome, and transcriptome during AA starvation (Fig 1F). Changes in the proteome were gradual across the time course reaching a maximum of >3,000 proteins significantly regulated by 10 h of starvation (Fig 1G). Using hierarchical clustering, we identified two main temporal clusters: ~ half the proteins decreased over time (enriched with ribosomal components), whereas another group of proteins increased, particularly proteins associated with lysosome biogenesis and function (Fig 1H–J). This pattern reflected a reduced need for ribosomal activity and an increased reliance on lysosome function, consistent with other results using murine ES cells (Todorova et al, 2024).

Within the phosphoproteome, we detected thousands of phosphorylation events (Fig 1I). Many of these modifications were linked to cell cycle control, particularly involving cyclin-dependent kinases (CDK), which is consistent with the shutdown of proliferation in cells exiting the cell cycle. At the mRNA level, we observed 8–10,000 transcripts were up- and down-regulated relative to unstarved cells (Fig 1G, right panel). Given the overall reduction of translation during AA starvation (Fig 1B), the transcriptomes and proteomes were largely uncorrelated (Fig S2A and B) This likely reflects a shift from bulk protein synthesis to the selective translation of stress-protective factors and the stabilization of specific proteins, alongside the activation of specialized transcriptional programs. Together, these findings illustrate that the adaptive response to long-term nutrient stress of single essential AAs is a multilevel process involving extensive transcriptional, translational, and post-translational reprogramming to achieve a temporary quiescent state.

GCN2 (encoded by Eif2ak4) is activated by limiting AA amounts. To understand the contribution of GCN2 to the broad rewiring of the transcriptome in Leu- or Arg-starved ES cells, we generated Eif2ak4-deficient ES cells by CRISPR/Cas9 targeting (Fig S1B). As expected, activation of phosphorylation of GCN2 at the key T889 site was induced by deprivation of Arg or Leu, followed by the activation of the ISR as measured by ATF4 expression (Fig S1C). We used the Eif2ak4-deficient ES to perform RNAseq in comparison with control ES cells. We observed that GCN2 (via its downstream mediators such as ATF4) controlled a substantial fraction of transcripts regulated by AA starvation, such as Txnip1 and Ddit3 (also known as CHOP), many of which are known targets of the integrated stress response controlled by GCN2 (Pakos-Zebrucka et al, 2016; Costa-Mattioli & Walter, 2020; Torrence et al, 2021) (Fig S1D and E). However, many classes of regulated transcripts were independent of GCN2, including lysosome-associated mRNAs and many transcripts encoding metabolic enzymes (Fig S1D). Therefore, starved ES cells rewired their transcriptional program, only part of which is GCN2-dependent, suggesting other pathways contribute to the overall AA stress response.

Although Arg and Leu are sensed by different upstream regulators—such as Sestrin-2 for Leu and Castor-1 for Arg—and are known to differentially affect ribosome pausing and the ISR (Darnell et al, 2018), our data from ES cells indicate a remarkably similar adaptive response to starvation of either AA. Quantitative comparisons of proteomic (10-h) and transcriptomic (3-h) data from Arg- or Leu-starved cells showed linear correlations (Fig S2A and B), a trend that extended to the phosphoproteome across all time points (Fig S2C) and to the specific proteins enriched during starvation (Fig S2D). These findings suggest that in ES cells, Arg and Leu deprivation converge on a common adaptive program. This finding contrasts with observations in transformed cells where Arg deprivation more robustly induces ribosome pausing and GCN2 activation compared with Leu starvation (Darnell et al, 2018).

### Metabolic adaptation in the quiescent state

Given that the transcriptome and proteome results showed thousands of changes in response to Arg or Leu starvation, we next considered how ES cells globally adapted their metabolism once

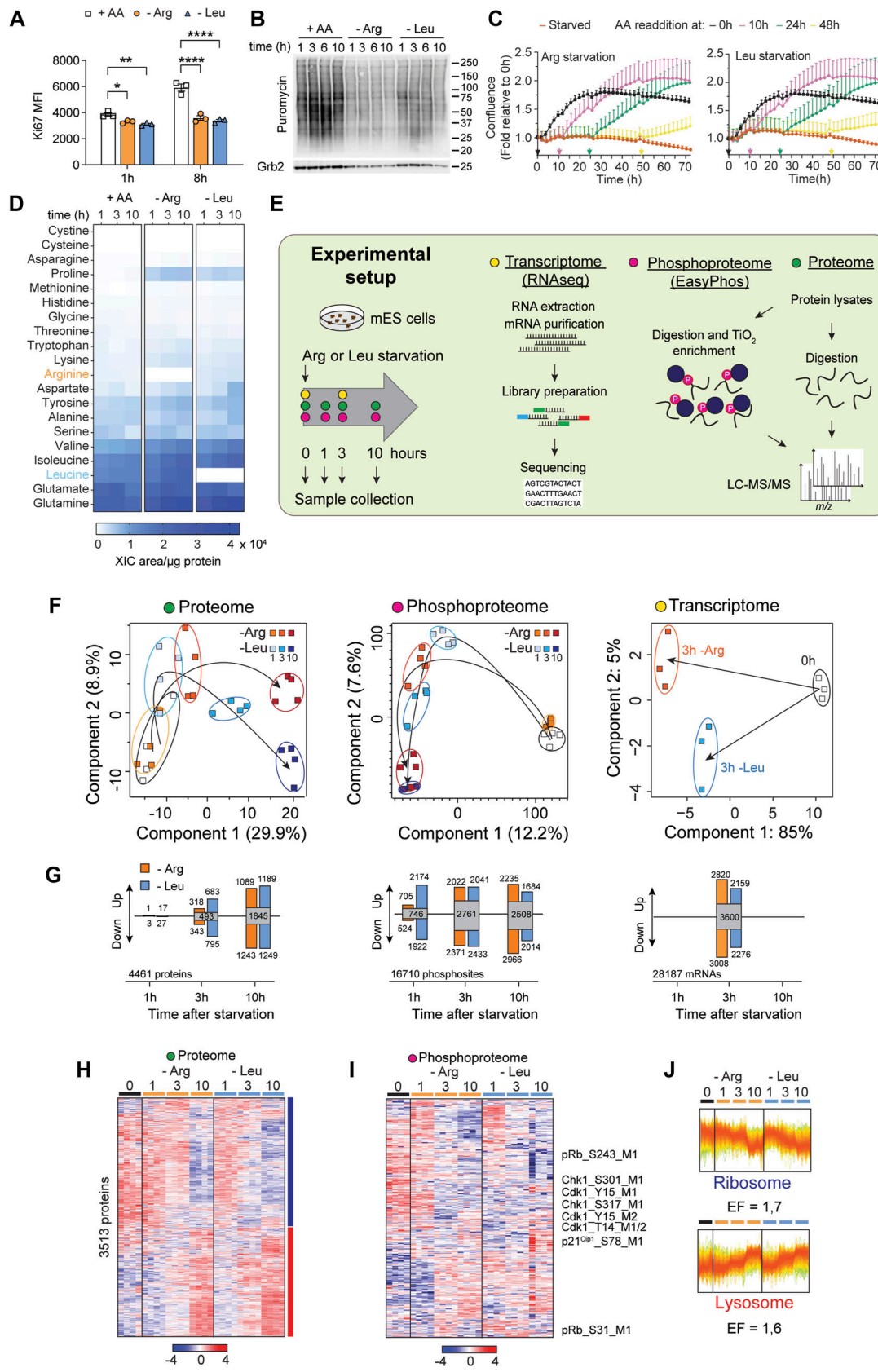

they entered a quiescent state. After an extended Arg or Leu starvation (17 h), the oxygen consumption rate remained near the assay baseline (Fig 2A) and the mitochondrial membrane potential (reflecting mitochondrial activity) was indistinguishable from that of proliferating cells (Fig 2B). Key metabolic intermediates (e.g., glucose-6-phosphate and succinate) were reduced (Fig 2C), and this paralleled a decline in the NADH/NAD$^+$ and NADPH/NADP$^+$ ratios (Fig 2D), whereas the ATP/ADP ratio increased (Fig 2E). In line with the transcriptomic and proteomic correlations, we did not observe any significant differences between Leu versus Arg starvation. These metabolic changes, together with preserved AA levels (Fig 1D) and rapid cell cycle reentry upon nutrient restoration (Fig 1C), suggest that ES cells adopt a metabolic state that is poised for rapid recovery. Interestingly, ~50% of key enzymes involved in central metabolism declined at the protein level. However, few corresponding mRNA changes (only 3/36) were detected (Fig 2F and G), reflecting the noncorrelation between the AA starvation transcriptome–proteome (Fig S3A and B). When considered collectively, the metabolic status of AA-starved cells was consistent with an overall decline in central metabolism, defined here by the pentose phosphate pathway, glycolysis, lactate production, and the TCA cycle.

## Cysteine or threonine deprivation causes distinct forms of cell death

We next considered whether we could correlate the proteomic, transcriptomic, and metabolic changes we quantified with Arg or Leu limitation to starvation of two other AAs needed for ES functionality beyond translation: Cys and Thr. Cys is required for glutathione (GSH) biosynthesis, translation, trans-sulfuration, Fe-S biogenesis, and other core redox-related cellular functions (Poltorack & Dixon, 2022; Zheng & Conrad, 2024). GSH is a core redox regulator essential for stress protection, including safeguarding against ferroptosis (Zheng & Conrad, 2024). Upon Cys removal, ES cells began to die after ~10 h (Fig 3A and B). Death caused by the absence of Cys was rescued by the addition of free radical scavengers ferrostatin-1 (Fer-1) and N-acetyl-cysteine (NAC) but not by inhibition of caspase-dependent apoptosis by

z-VAD-fmk or necroptosis by Nec1s (Fig 3A and B), indicating that Cys starvation induced ferroptosis in ES cells.

Thr use is a specific and essential component of murine ES cell metabolism and is controlled by the rate-limiting enzyme threonine dehydrogenase (THD), which is not expressed in primate ES cells (and is a silenced gene in humans) (Wang et al, 2009). In contrast to the reversible quiescence induced by Arg or Leu deprivation, Thr starvation led to complete cell death by 24 h (Wang et al, 2009) (Fig 3C). Thr starvation–induced death took ~8 h to initiate as readdition of Thr at the 8-h time point rescued cell survival. However, this form of AA starvation death was not rescued by inhibitors of apoptosis, necroptosis, or ferroptosis, suggesting the involvement of an alternative, nonconventional death pathway we term threonine deprivation–associated cell death (Fig 3D). Thus, starvation of different AAs initiated distinct changes in cellular responses: Arg and Leu deprivation enforce cellular quiescence, whereas -Cys primes ES cells for ferroptosis and Thr starvation causes a nonconventional model of cell death.

## Cross-talk between AA starvation pathways

Previous results have shown that cell cycle arrest protects cells from ferroptosis, in part by preserving GSH levels (Tarangelo et al, 2022). We therefore hypothesized that deprivation of Arg or Leu, a process that enforces cell cycle exit, would suppress the lethal effects of Cys or Thr withdrawal. To test this idea, we simultaneously withdrew Arg or Leu in combination with Cys (i.e., -Cys, -Arg or -Cys, -Leu). Both "double" starvations partly blocked Cys starvation–induced ferroptosis including lipid peroxidation (Figs 3E and S4A–C). However, Cys-starved ES cells eventually died (Fig 3E), highlighting the essential nature of exogenous Cys even in quiescent cells. In contrast, withdrawal of Thr coincident with Arg or Leu could not rescue the cells (Fig 3F and G). These findings further emphasize that "threonine deprivation–associated cell death" was the distinct form of cell death. To investigate the effects of Thr starvation death further, we next performed a "screen" to determine whether withdrawal of essential AAs beyond Arg and Leu would influence threonine deprivation–associated cell death, and we prepared media lacking Thr combined with 13 other AAs

**Figure 1. Time-resolved multi-omics profiling of the AA starvation response.**
**(A)** ES cells were starved of Arg or Leu, and cell proliferation was assessed by Ki67 expression and intracellular flow cytometry. As a control, medium containing all AAs was used. Data are shown as the mean of fluorescence intensity (MFI) ± SEM, n = 3. Statistical significance was determined by two-way ANOVA followed by Dunnett's multiple comparisons: *$P$ < 0.05, **$P$ < 0.01, ****$P$ < 0.0001. **(B)** Newly synthesized proteins were labeled by puromycin incorporation and revealed by immunoblotting. **(C)** Cell proliferation during Arg or Leu deprivation and resupplementation at the indicated time points (arrows). Cell confluence was monitored by live imaging and normalized to 0 h, and is shown as the mean ± SEM, n = 3. **(D)** Heatmap of all amino acids detected by LC-MS–based metabolomics in the cell lysates at 1, 3, and 10 h after Arg or Leu starvation. The data were calculated as the mean of the extracted ion chromatogram (XIC) per area of three replicates normalized to the total amount of protein. **(E)** Schematic of the experimental setup and workflow of the omics data analysis. **(F)** Principal component analysis assessing temporal dynamics during Arg (orange, red shades) and Leu (blue shades) starvation. Each square represents a sample collected at a particular time point as indicated, with lighter and darker shades denoting early and late time points, respectively. **(G)** Summary statistics of proteins, phosphosites, and transcripts (read counts > 1) significantly up- or down-regulated at the indicated time points during Arg and Leu starvation, as determined by multiple $t$ tests relative to 0 h (proteome, phosphoproteome, and FDR < 5%; transcriptome: Wald's test, Benjamini–Hochberg-adjusted $P$-value [FDR] < 5%). Gray boxes indicate common proteins, phosphoproteins, or transcripts between Arg and Leu starvation. **(H, I)** Heatmap of z-scored intensities of the proteome (H) and phosphoproteome (I). **(H)** Proteome was analyzed by hierarchical clustering (Euclidean distance, y-axis) of all significantly regulated proteins (3,513) compared with all identified (4,461) (one-way ANOVA, $s0$ = 0.1, FDR < 5%). Profiles of each cluster based on protein temporal dynamics are color-coded according to their distance from the respective cluster center as shown on the top in hr of starvation. **(I)** Phosphoproteome was analyzed by hierarchical cluster analysis (Euclidean distance) and represents all significantly regulated sites (8,715) (one-way ANOVA, $s0$ = 0.1, FDR < 5%). **(J)** Fisher's exact test on protein clusters with ribosome and lysosome KEGG annotations (background is all identified proteins in the dataset, cutoff $P$ < 0.002). The enrichment factors (EF) are displayed. **(H)** Top (ribosome) and bottom (lysosome) panels correspond to blue and red clusters in (H), respectively.
Source data are available for this figure.

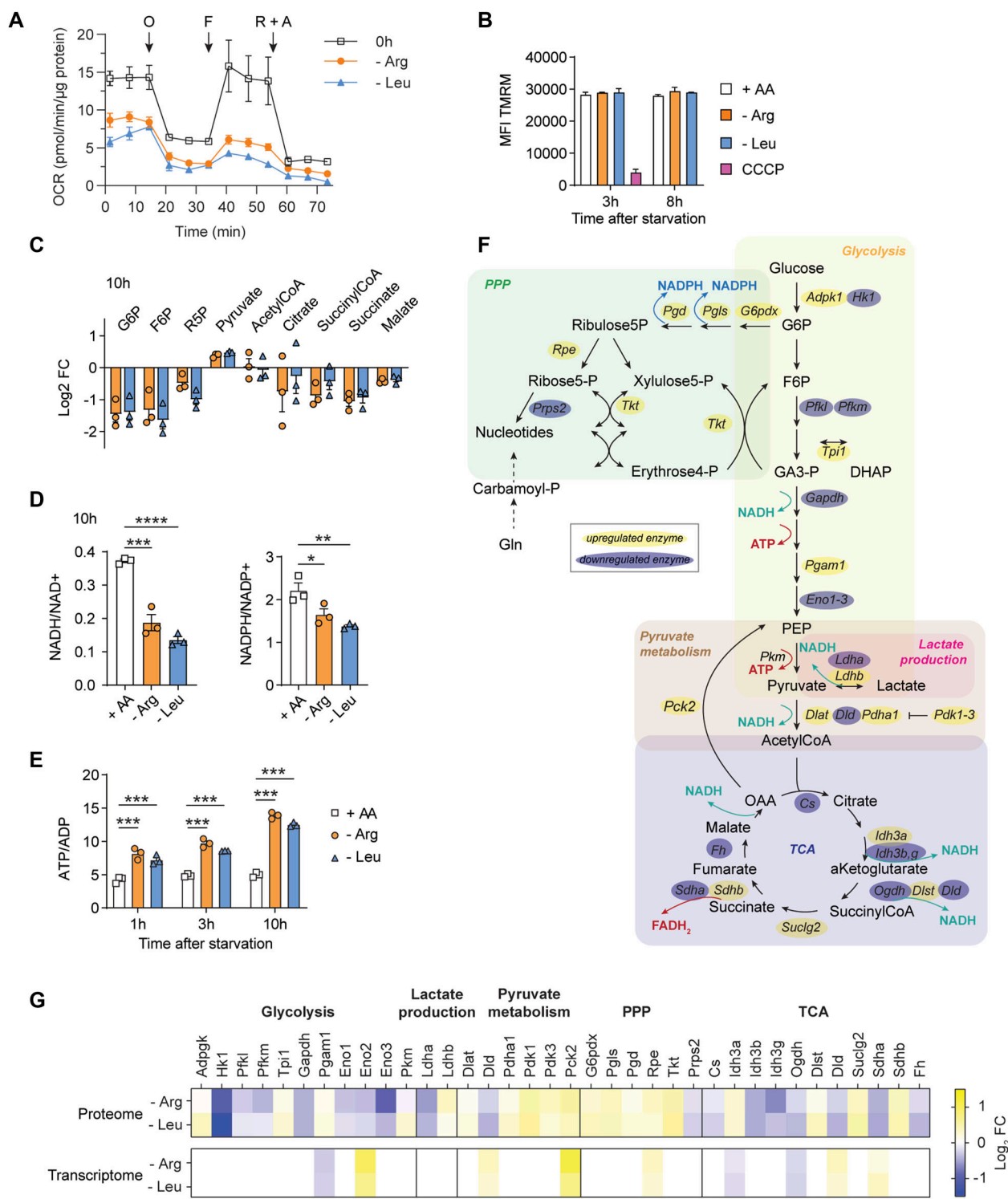

**Figure 2. Metabolic adaptation during the amino acid starvation response.**
**(A)** Mitochondrial respiration of cells is reduced during prolonged amino acid deprivation. Cells were starved of Arg or Leu overnight (17 h) and assessed by Seahorse. Oxygen consumption rate measurements were normalized to the amount of protein in the samples and are shown as the mean ± SEM of three replicates. O, oligomycin; F, FCCP; R+A, antimycin A + rotenone. **(B)** Mitochondrial potential is unaffected during Arg or Leu limitation, as assessed by the TMRM fluorescent probe. As a control, 50 μM CCCP was added to inhibit oxidative phosphorylation. The results display the mean ± SEM of a representative experiment with two replicates. **(C, D, E)** HPLC-MS–based metabolomics revealed a decline in central metabolism after Arg and Leu starvation. **(C)** Changes in the abundance of the main metabolic intermediates 10 h after starvation are displayed as the mean of the fold change of Arg- or Leu- samples compared with the unstarved (+AA) group (n = 3) (C). **(D, E)** NADH/NAD+, NADPH/NADP+ (D), and ATP/ADP (E) ratios are shown as the mean ± SEM of three replicates and calculated at 10 h (D) or the indicated time points (E) after Arg or Leu starvation. Significant differences were obtained by one-way or two-way ANOVA followed by Dunnett's multiple comparisons at each time point: *P < 0.05, **P < 0.01, ****P < 0.0001. **(F)** Scheme of the metabolic pathways summarizing the metabolic and related proteomic changes. The metabolic enzymes are highlighted according to the proteome dataset: in

individually dropped out (Fig 3H and I). Using this pairwise approach, only withdrawal of Met rescued cell death.

Arg or Leu starvation produced a highly transcriptional and proteomic response, including similar activation of the GCN2-ISR (Figs S1E, S2, and S3). We therefore wondered whether lack of Thr or Cys would also trigger similar "generic" AA starvation responses, even though the final fate of deprivation of either AA was a distinct form of cell death. We therefore investigated the transcriptional responses to Thr or Cys starvation. Surprisingly, Thr starvation massively altered the ES transcriptome with thousands of transcripts altered relative to Arg or Leu starvation combined with a heightened ISR (Fig S5A). However, genetic loss of GCN2 or inhibition of GCN2 with GCN2-IN-6 did not influence the outcome of pairwise AA dropouts and -Met rescue of threonine deprivation–associated cell death (Fig S6A–C). In contrast to Thr, the absence of Cys triggered far fewer transcriptional changes compared with either -Thr or -Arg/-Leu conditions (Fig S5B). Surprisingly, the quality of the transcriptional responses was also distinct between -Cys, -Thr, and -Arg/-Leu as estimated by a ranking factor metric we used to compare different RNAseq datasets (see the Materials and Methods section) (Fig S5C–E). Collectively, these results emphasize that ES cells deploy different strategies to negotiate the absence of different AAs.

### Cell cycle exit and global translational suppression confer stress protection

The effects of Met starvation on threonine deprivation–associated cell death (Fig 3H and I) suggest two mutually nonexclusive possibilities to account for the death suppression: first, the absence of Met could alter the Thr catabolic pathway. Indeed, in ES cells both AAs are linked via the 1-carbon and folate cycles (Wang et al, 2009). Second, the absence of Met could broadly affect translation at both the initiation and elongation steps. We noted that inhibition of mTOR via torin-1 but not rapamycin, which has a different mode of action (Thoreen et al, 2009; Kang et al, 2013; Shimobayashi & Hall, 2014; Bohm et al, 2021) on mTOR complexes, triggers cell cycle exit, quiescence, and translation inhibition similar to the absence of Leu or Arg (Figs 4A and B and 1B) and therefore tested the latter possibility, namely, that -Met mimics translation inhibition and blocks threonine deprivation–associated cell death. We compared -Met with the addition of torin-1 to Thr-starved ES cells. Remarkably, torin-1 completely rescued threonine deprivation–associated cell death (Fig 4C), which was confirmed by experiments to block mTOR enzymatic activity (sapanisertib and rapalink), whereas rapamycin had no effect (Fig S6D). If translation was the critical step in this process, then a broad translation inhibitor such as cycloheximide (CHX) should produce the same effect. Indeed, CHX also blocked threonine deprivation–associated cell death (Fig 4D and E). These results suggest that Thr deprivation triggers the synthesis of a protein(s) that executes non-apoptotic,

non-necropotic, or non-ferroptotic death in a Met- and mTOR-dependent manner.

We next investigated the effects of translation inhibition on Cys starvation. We therefore hypothesized that processes that enforce cell cycle exit via translational inhibition would also block ferroptosis in ES cells. Consistent with this idea, CHX or anisomycin also protected ES cells from Cys starvation–induced cell death (Fig 4F). Collectively, therefore, cell cycle exit and translational suppression were sufficient to protect ES cells from the lethal effects of Cys withdrawal, providing a common broad protective pathway for blocking death by Cys or Thr starvation.

An emerging concept in cancer therapy is that "persister" cells that survive chemotherapy are forced, temporarily, from the cell cycle and remain in a quiescent-like state as a survival strategy (Hangauer et al, 2017; Takahashi et al, 2020; Oren et al, 2021; Pu et al, 2023). In tumor microenvironments, amino acids such as arginine and tryptophan can be limiting (Sullivan et al, 2019), and as we found here for Arg and Leu, a quiescent-like state is enforced when these AAs are deprived. We therefore wondered whether Arg or Leu depletion could confer broad stress protection to ES cells because starvation of these AAs enforces reversible cellular quiescence, unlike Cys or Thr starvation, which triggers cell death. We starved ES cells or Arg or Leu in combination with inhibitors of vital metabolic checkpoints including GSH synthesis, NADPH production, and 1-carbon metabolism (Gclc, Cth, G6pdx, Mthfd2, or Shmt2 using L-buthionine-sulfoximine (BSO), DL-propargylglycine (PAG), 6-aminonicotinamide (6AN), DS18561882 (DS), or SHIN1, respectively). We used -Cys as an internal control for ES cell death that is partly blocked by Arg or Leu withdrawal (Fig 3E and F). As expected, all of these inhibitors were highly toxic to proliferating ES cells. However, ES cells deprived of Arg or Leu were differentially protected from the cytotoxic effects of each metabolic inhibitor (Fig 4G). These results indicate that Arg or Leu deprivation, via cellular quiescence, provides partial protection against the effects of disruption of key metabolic nodes.

## Discussion

Most of the research on mammalian amino acid sensing has focused on how cells integrate amino acid availability through mTORC1, a major anabolic signaling hub. Most studies have used tractable, transformed cell models, such as HeLa and 293T, which rapidly respond to amino acid withdrawal by down-regulating mTORC1 signaling, thereby halting anabolic growth. Further studies in various transformed cell types revealed that the routes for acquiring amino acids to support malignant growth are elastic (Saxton & Sabatini, 2017; Vettore et al, 2020; Shorthouse et al, 2022). These cells can scavenge amino acids from extracellular sources like serum and matrix (Commisso et al, 2013; Sousa et al, 2016; Nofal et al, 2017), by altering the expression of SLC transporters and rewiring intrinsic pathways, including autophagy, lysosomal

yellow or blue whether they are up-regulated or down-regulated after 10 h of Arg or Leu starvation compared with 0 h, respectively. **(G)** Heatmap of the proteome and transcriptome changes ($\log_2$ fold change compared with the unstarved group) associated with metabolic adaptation. Source data are available for this figure.

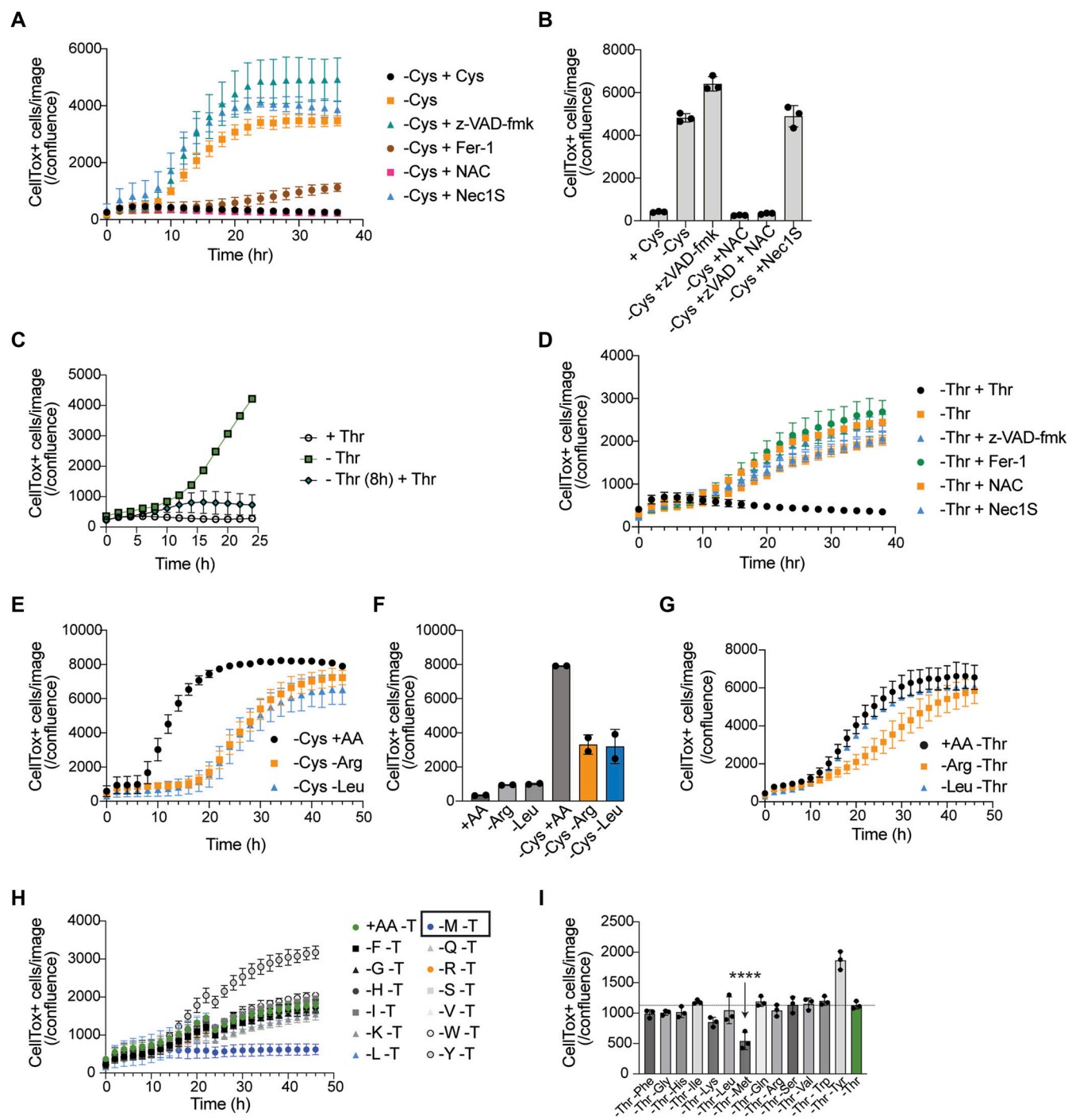

**Figure 3. ES cells are sensitive to Cys withdrawal.**
**(A)** Quantification of cell death induced by Cys deprivation in the presence or absence of cell death inhibitors. Note that death induced by Cys starvation is blocked by readdition of Cys, Fer-1, or NAC. A representative experiment with two technical replicates is shown. The data were calculated as a ratio of CellTox⁺ green object counts per image to the cell confluence area. **(B)** Summary quantification of cell death induced by Cys deprivation in the presence or absence of cell death inhibitors at 24 h for each condition. ****$P < 0.0001$, Dunnett's multiple comparison test after ordinary one-way ANOVA. **(C)** -Thr–induced cell death can be rescued by readdition of Thr at 8 h post-starvation. Cell death (CellTox+ cells/confluence) was measured over time. **(D)** Time course of cell death triggered by Thr starvation in the presence of z-VAD-fmk, Fer-1, NAC, or Nec1s. Note that only the readdition of Thr can rescue cell death. Cell death (CellTox⁺ cells/confluence) was measured over time by IncuCyte imaging. **(E)** Arg- or Leu-starved cells are partly protected from the effects of Cys deprivation. Cells were deprived of Cys (black dots) or concurrent starvation with -Arg or -Leu, and cell death (CellTox⁺ cells/confluence) was measured over time. The data shown are representative of >10 experiments. **(F)** Summary data (24 h) of the effect of Arg- or Leu-starved cells on Cys starvation–induced death from two independent experiments. **(G)** ES cells were starved of Thr with concomitant withdrawal of Arg or Leu, and cell death (CellTox⁺ cells/confluence) was measured over time. **(H)** ES cells were starved of Thr with concomitant withdrawal of each AA shown, and cell death (CellTox⁺ cells/confluence) was measured over time. Only concomitant loss of Met rescues cell death (boxed). **(I)** ES cells were starved of Thr with concomitant withdrawal of each AA shown and cell death (CellTox⁺ cells/confluence) assessed at 24 h (n = 3). ****$P < 0.0001$, Dunnett's multiple comparison test after ordinary one-way ANOVA relative to all sample groups except -Thr -Tyr.

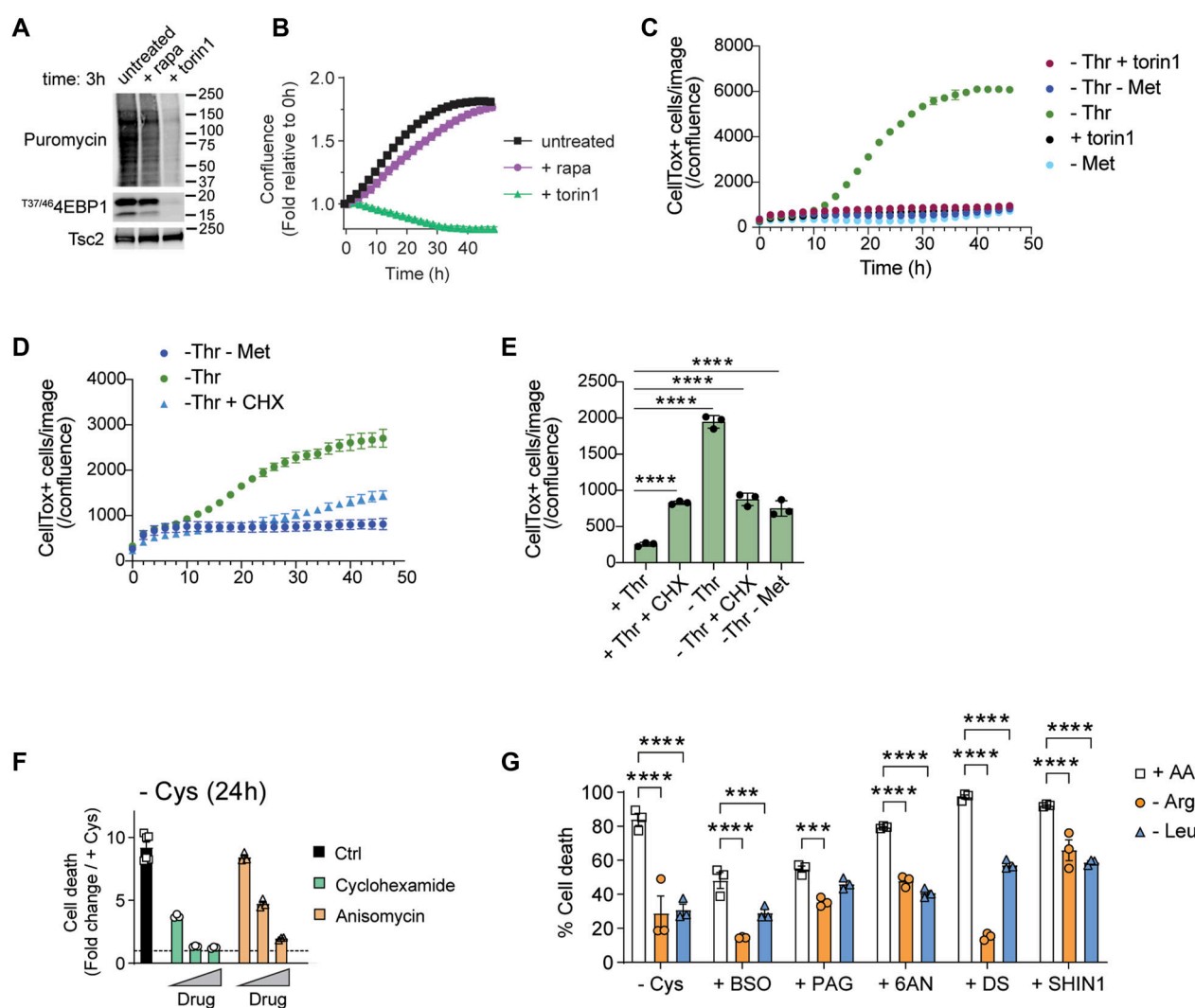

**Figure 4. Translational perturbation mediates protection to cell stress.**
**(A)** Immunoblotting of translation and mTOR activity 3 h after treatment with mTOR inhibitors. Nascent proteins were labeled by puromycin. Tsc2 was used as a loading control. **(B)** Cell proliferation in the presence of 500 nM rapamycin and torin1. Cell confluence was monitored by live imaging and normalized to 0 h, and is shown as the mean ± SEM of three replicates. **(C)** ES cells were starved of Thr in the presence of torin-1 or concomitant withdrawal of Met, and cell death (CellTox$^+$ cells/confluence) was measured over time. **(D)** ES cells were starved of Thr in the presence of cycloheximide (CHX), or concomitant withdrawal of Met, and cell death (CellTox$^+$ cells/confluence) was measured over time. **(E)** Summary data (24 h) of the effect of Met withdrawal or CHX addition on Thr starvation–induced death (n = 3). ****$P < 0.0001$, Dunnett's multiple comparison test after ordinary one-way ANOVA relative to the +Thr control group. **(F)** Cys-starved cells are partly protected from ferroptosis by translational inhibitors CHX and anisomycin. Shown are normalized data to untreated controls (1, dotted line). The data shown are representative of >10 experiments. **(G)** Cells were subjected to the following treatment for 24 h during Arg or Leu limitation: Cys deprivation, 1 mM BSO, 75 $\mu$M 6AN, 2 mM PAG, 20 $\mu$M DS, or 75 $\mu$M SHIN1. Living cells were measured by flow cytometry as Annexin V-/7AAD-, and the percentage of cell death was calculated as the remaining cells, and is shown as the mean ± SEM of three replicates. **(D)** Significant differences were obtained by two-way (D) ANOVA followed by Dunnett's multiple comparisons: *$P < 0.05$, **$P < 0.01$, ***$P < 0.001$, ****$P < 0.0001$.
Source data are available for this figure.

functions, and catabolism of abundant protein complexes such as ribosomes. Moreover, the ease of manipulating amino acid signaling in these models has facilitated the identification of key amino acid–dependent upstream regulators of mTORC1, such as the GATOR complexes and their leucine and arginine regulators, Sestrin-2 and Castor-1, respectively (Bar-Peled et al, 2013; Wolfson et al, 2016; Wolfson & Sabatini, 2017; Costa-Mattioli & Walter, 2020).

In contrast, our study set out to map the signaling events and adaptive pathways elicited by amino acid starvation in a non-

transformed cell system, aiming to understand AA signaling and adaptation in the absence of oncogene-driven pathways. We chose murine ES cells for their rapid cell division rate and experimental reproducibility. Upon deprivation of Leu or Arg, ES cells exit the cell cycle, reduce translation, and become resistant to diverse stresses, including perturbations of glutathione and 1-carbon metabolism, which are essential for rapidly dividing cells. In contrast, when deprived of Cys or Thr, ES cells undergo cell death via distinct mechanisms: Cys deprivation induces ferroptosis, whereas Thr

deprivation triggers a form of cell death that is neither apoptosis, necroptosis, nor ferroptosis. Notably, threonine deprivation–associated cell death can be rescued by co-removal of Met, or by treatment with torin-1 or other broad translation inhibitors, highlighting three distinct cellular outcomes driven by the deprivation of four amino acids. One possibility to account for the protective effects of Met withdrawal or translation inhibitors on threonine deprivation–associated cell death concerns the possibility that a newly synthesized protein(s) is necessary for the elicitation of the death pathway. Although these results suggest a mTORC1-dependent process such as 4EBP phosphorylation and subsequent CAP-dependent translation initiation is involved in the synthesis of a new protein(s), at this point we cannot definitively rule out a role for mTORC2 in threonine deprivation–associated cell death because our results thus far are based on the use of chemical inhibitors and global manipulation of AA balance. Finding such a protein(s) will require a genetic-based screen and may give general insights into unexplored death pathways that intersect with core cellular metabolism.

A notable difference between the death responses to Cys versus Thr starvation was that coincident starvation with either Arg or Leu was partly protective against -Cys ferroptosis but did not protect or even slightly enhanced threonine deprivation–associated cell death. One explanation for this phenomenon may be linked to the rate of intersection of pathways that mobilize reserve AAs. In the case of -Leu or -Arg on partial suppression of ferroptosis caused by Cys starvation, death is delayed by ~10 h, whereas no effects occurred on threonine deprivation–associated cell death. Perhaps Arg or Leu starvation coincidentally triggers the mobilization of Cys. In contrast, insufficient Thr is available. These speculative ideas require further experimentation and reinforce the notion that our understanding of the effects of AA starvation is limited.

We have limited information about how normal cells negotiate temporary restrictions in amino acids. Herein, we used the term "normal" cells as non-transformed. Furthermore, the many types of "normal" cells in the body likely have vastly different requirements for amino acid supplies. For example, neurons are mainly nondividing cells but have large cell masses and substantial metabolic demand, and use vast amounts of some amino acids such as neurotransmitter precursors (e.g., glycine, glutamine, tryptophan, and tyrosine). Similarly, intestinal epithelial cells, which continuously regenerate in crypts, compete with the microbiota for nutrients, including amino acids. T cells, upon activation by antigen, rapidly alter their amino acid transporter profiles while decoupling amino acid supply from mTORC1, which responds to T-cell receptor and co-stimulatory signaling (Hukelmann et al, 2016; Van de Velde & Murray, 2016; Van de Velde et al, 2017; Howden et al, 2019; Marchingo et al, 2020). These examples underscore the varied amino acid needs that different cell types encounter in vivo.

Our findings also indicate that forcing ES cells to exit the cell cycle is sufficient to confer protection against different metabolic stress perturbations, including redox stress. Our results are consistent with recent studies in transformed cell systems where broad redox stress protection is afforded by cell cycle exit, which reroutes metabolic pathways away from translation and anabolic metabolism and toward resource conservation (Conlon et al, 2021;

Tarangelo et al, 2022). It is tempting to speculate that AA limitation may be inherently linked to cellular stress protection. In tumor microenvironments, selective AAs are limiting (Sullivan et al, 2019) and arginine- and tryptophan-degrading enzymes are expressed and associated with tumor survival (Murray, 2016; Zeitler & Murray, 2023), which is generally thought to suppress T-cell proliferation and function. In complex environments like tumors, AA starvation could affect cellular physiology in a differential way: rapidly proliferating cells, including malignant cells, might be forced out of the cell cycle by neighboring cells expressing AA-degrading enzymes (combined with the effects of local nutrient consumption and reduced perfusion in solid tumors). This adaptation may promote survival of some transformed cells through rewiring of their AA response pathways. Although speculative, AA stress may protect against other forms of stress in a complex in vivo milieu.

Our study has several limitations. First, ideally, a comprehensive, multiparametric comparison of the responses to starvation by all 20 amino acids in both non-transformed and transformed cells over time would reveal distinct adaptation classes and highlight specific pathways involved in amino acid sensing and response. Such an approach should reveal "classes" of adaptation and point the way to specific pathways necessary for the detection of an AA and the corresponding adaptation to the loss of the AA. However, although the experimental scale of such an approach is feasible, the analysis of outcomes would be highly challenging. Our study showed thousands of changes in the proteomes, phosphoproteomes, and transcriptomes of Arg- and Leu-starved ES cells, yet the full physiological implications of these changes remain only partially understood. Furthermore, recent findings implicate different AA paths to control the subcellular distribution and activity of mTORC1. For instance, Leu and Arg appear to regulate the lysosomal fraction of mTORC1 (e.g., via the GATOR complex), whereas Cys and Thr regulate non-lysosomal mTORC1 complexes (Fernandes et al, 2024). The astonishing complexity of the pathways that integrate AA quantities to downstream signaling networks suggests that cells have precise ways to detect different AAs, which remain largely unknown. Second, we did not use methods to label nascent proteins during amino acid stress (An et al, 2020; Klann et al, 2020). Although translation slows after cell cycle exit, it does not cease entirely, as amino acids can still be obtained from cellular sinks even during prolonged starvation. Consequently, the identity of the proteins synthesized during the complete absence of an exogenous AA remains unclear, beyond several key targets of the ISR, such as ATF4 and CHOP. Future studies using unnatural AAs as labeling tools will be essential to identify and quantify nascent protein biosynthesis during amino acid limitation, potentially uncovering novel targets for addressing cancer cell persistence.

## Materials and Methods

### Cell culture

E14 mouse ES cells were a gift from Prof. D. Nedialkova (MPIB) and were originally purchased from the American Type Culture Collection (CRL-1821; ATCC). ES cells were grown on 0.1% gelatin-

coated dishes in DMEM high-glucose GlutaMAX containing 15% FBS (Life Technologies), penicillin–streptomycin, $\beta$-mercaptoethanol, and leukemia inhibitory factor (murine recombinant LIF, 10 ng/ml). All cell culture reagents were from Thermo Fisher Scientific unless otherwise specified. All cell lines were grown in humidified tissue culture incubators at 37°C with 5% $CO_2$. Cells were routinely tested to be free of *Mycoplasma* contamination by PCR testing (LookOut *Mycoplasma* PCR Detection Kit, MP0035; Sigma-Aldrich).

## Single AA starvation experiments

$0.5 \times 10^6$ ES cells were seeded in 12-well plates for 12 h. Before starvation, cells were washed 3 times with PBS to remove any AA left. In Arg and Leu starvation experiments, we used SILAC DMEM (D9443; Sigma-Aldrich) supplemented with penicillin–streptomycin, GlutaMAX, glucose, 5% dialyzed FBS (dialyzed in-house), $\beta$-mercaptoethanol, LIF, and L-Lys, and with or without Arg or Leu. L-Gln starvation was performed in DMEM (D9800-13; US Biological) containing penicillin–streptomycin, 3.7 g/liter sodium bicarbonate, 4.5 g/liter glucose, 5% dialyzed FBS, $\beta$-mercaptoethanol, and LIF, and supplemented with L-Cys, Gly, L-His, L-Iso, Leu, L-Lys, L-Met, L-Phe, L-Ser, L-Trp, L-Tyr, L-Val, and either Arg, Leu, or L-Gln as indicated in the figure. AAs were from Carl Roth and made as 200X (Arg, L-His, L-Iso, Leu, L-Lys, L-Phe, L-Trp, L-Val) or 1000X (L-Cys, Gly, L-Met, L-Ser) stocks in PBS, directly resuspended in medium (L-Tyr), or obtained as a ready-made solution (L-Gln, 100X; Life Technologies).

## Drugs and other reagents

The following compounds were used as indicated in the figure legend: 6-aminonicotinamide (6AN, A68203; Sigma-Aldrich), anisomycin (A9789; Sigma-Aldrich), L-buthionine-sulfoximine (BSO, B2515; Sigma-Aldrich, 200 mM stock resuspended in water), cycloheximide (C7686; Sigma-Aldrich), DS18561882 (DS, HY-130251; MedChemExpress), GCN2-IN-6 (resuspended in DMSO to 10 mM and used as previously described [Bruggenthies et al, 2022]), MG132 (474790; Millipore), mitomycin C (M4287; Sigma-Aldrich, resuspended in water), DL-propargylglycine (PAG, P78888; Sigma-Aldrich, 400 mM stock resuspended in water), 1 $\mu$M RSL3 (S8155; SelleckChem), SHIN1 (HY-112066; MedChemExpress). Dinaciclib (HY-10492; MedChemExpress) and palbociclib (S1116; SelleckChem, resuspended in water) were used to inhibit CDK1/2/5/9 and CDK4/6, respectively. 2 $\mu$M Ferrostatin-1 (Fer-1, SML0583; Sigma-Aldrich), 10 $\mu$M z-VAD-fmk (MedChemExpress in DMSO 10 mM), and 10 $\mu$M Nec1S (Cell Signaling Technologies in DMSO 10 mM) were used to inhibit ferroptosis, apoptosis, and necroptosis, respectively. mTOR was directly inhibited using rapamycin (533210; Millipore) or torin-1 (14379; Cell Signaling). All drugs and compounds were resuspended in DMSO unless otherwise specified.

## CRISPR/Cas9

To generate an *Eif2ak4*$^{-/-}$ genetic deficiency in ES cells, sgRNA sequences were designed using standard design tools (IDT) and cloned into the pX458 CRISPR vector (pSpCas9(BB)-2A-GFP, #48138; Addgene). Single-cell cloning was performed directly after sorting

for GFP$^+$ cells in 96-well plates, which were then selected based on the target protein expression and/or sequencing. The following sgRNA and crRNA sequences were used:

*Eif2ak4* sgRNA_1 (exon 9): GCAAAGTAGCGGACGATATT.
*Eif2ak4* sgRNA_2 (exon 9): AGTAGCGGACGATATTTGGA.

## Immunoblotting

Cell lysates were prepared on ice in RIPA buffer (ab156034; Abcam) containing protease and phosphatase inhibitors (78444; Thermo Fisher Scientific). As applicable, protein concentrations were determined using Pierce BCA Protein Assay Kit (23225; Thermo Fisher Scientific) and a NanoDrop 2000 spectrophotometer (Thermo Fisher Scientific). Proteins were separated on 4–15% Criterion TGX Stain-Free Protein Gel (5678084/5; Bio-Rad) and transferred to 0.2 $\mu$M nitrocellulose (Amersham). Membranes were blocked in 1–3% BSA (A2153; Sigma-Aldrich) or 3% nonfat milk (T145.2; Carl Roth) in TBS/0.01% Tween-20 and probed overnight at 4°C with primary antibodies listed below. Membranes were washed and probed with secondary antibodies and developed using Super-Signal West Pico PLUS Chemiluminescent reagent (34580; Thermo Fisher Scientific). Homogeneous loading was controlled by blotting for vinculin, Grb2, or Tsc2.

## Antibodies

The following antibodies were used in immunoblotting experiments: phospho-4EBP1 (T37/46, 1:1,000, 2855; Cell Signaling), Atf4 (1:1,000, sc-390063; Santa Cruz), phospho-Gcn2 (T899, ab75836, 1:1,000; Abcam), Grb2 (1:1,000, 610112; BD Biosciences), puromycin (1:10,000, MABE343; Millipore), phospho-S6K (T389, 1:1,000, 9234; Cell Signaling), Tsc2 (1:1,000, 4308; Cell Signaling), vinculin (1:2,000, 66305-1-Ig; Proteintech).

## Estimation of mRNA translation

To estimate translation rates, we used surface sensing of translation (SUnSET) method (Schmidt et al, 2009) where puromycin incorporation in neosynthesized proteins approximates the rate of mRNA translation at a given time point. 10 $\mu$g/ml of puromycin (P8833; Sigma-Aldrich, 10 mg/ml stock resuspended in PBS) was added to the culture 10 min before sample collection. Samples were then subjected to Western blotting. When indicated, the following drugs were used to inhibit translation: cycloheximide (C7698; Sigma-Aldrich).

## Cell proliferation and cell death analyses

E14 ES cells were seeded in 48-well plates and treated as specified in the figure legend. Non-pharmacological ferroptosis induction was performed using L-Cys–free DMEM (D9800-13; US Biological) containing penicillin–streptomycin, 3.7 g/liter sodium bicarbonate, 4.5 g/liter glucose, 10% dialyzed FBS, $\beta$-mercaptoethanol, LIF, and the relevant AAs as described above. To evaluate proliferation, cells were fixed and permeabilized using BD Cytofix/Cytoperm and stained with anti-Ki67 FITC antibody (652410; BioLegend). All reagents were from BD Biosciences unless specified. All

measurements were recorded on the LSRFortessa with FACSDiva software (BD Biosciences) and further analyzed using FlowJo 10 software.

### Live microscopy

Data acquisition and analysis were performed using IncuCyte S3 Live Cell Analysis System (Sartorius). Nine images per replicate were taken every 2 h for 48 h using the 10X objective. Cell proliferation was evaluated by analyzing the cell confluence in the phase channel. For cell death measurements, CellTox Green (G8731; Promega) was used according to the manufacturer's instructions to stain cells with impaired membrane integrity. CellTox[+] Green object numbers were then counted in the fluorescence channel and normalized to the cell confluence for each well. The results are shown as a percentage of cell death after further normalization against the unstarved group in each dataset using 0% as cell death at the 0-h time point and 100% as the maximum cell death seen during the time course.

## Measurement of ROS production, lipid peroxidation, and mitochondrial potential

ROS production and lipid peroxidation were assessed according to the manufacturer's instructions. Briefly, 10 $\mu$M of CM-H$_2$DCFDA (C6827; Thermo Fisher Scientific) or 2 $\mu$M of C11-BODIPY 581/591 (D3861; Thermo Fisher Scientific) or 20 nM of tetramethylrhodamine methyl ester (TMRM, M20036; Thermo Fisher Scientific) was added to the cells 30 min before the end of the experiment. 20 $\mu$M of carbonyl cyanide 3-chlorophenylhydrazone (CCCP) was added to the cells for 5 min to induce mitochondrial membrane depolarization as a control. The MFI in the samples was measured by flow cytometry (LSRFortessa) and analyzed using FlowJo 10 software.

## RNAseq

Cells were starved of Arg, Leu versus control unstarved ES cells, or Cys or Thr versus unstarved cells in an independent experiment set, all in triplicates. RNA was extracted using Direct-zol RNA Miniprep Plus Kit (R2071; Zymo Research) according to the manufacturer's protocol. mRNA sequencing libraries were prepared with 1 $\mu$g of total RNA of each sample using the NEBNext Ultra II Directional RNA Library Prep Kit for Illumina (E7765; NEB) with NEBNext Poly(A) mRNA Magnetic Isolation Module (E7490; NEB), according to the standard manufacturer's protocol. Total RNA and the final library quality controls were performed using Qubit Flex Fluorometer (Q33327; Thermo Fisher Scientific) and 2100 Bioanalyzer Instrument (G2939BA; Agilent) before and after library preparation. Paired-end sequencing was performed on Illumina NextSeq 500 (2 × 43 bp reads). The samples were multiplexed and sequenced on one High Output Kit v2.5 to reduce batch effects. BCL raw data were converted to FASTQ data and demultiplexed by bcl2fastq Conversion Software (Illumina). After checking the quality of the samples (FastQC, v.0.11.7), the files were mapped to the mouse genome (Genome build GRCm38) downloaded from Ensembl using the star aligner (Dobin et al, 2013), version 2.7.4a. The mapped files were then quantified on a

gene level based on the Ensembl annotations, using the HTSeq tool (Anders et al, 2015) in Python. Using the DESeq2 package (Anders & Huber, 2010) (R 4.0.3, DESeq version 1.30.1), the count data were normalized by the size factor to estimate the effective library size. A filtering step of removing genes with less than 1 reads was used. This was followed by the calculation of gene dispersion across all samples. The analysis of two different conditions against each other resulted in a list of differentially expressed genes for each comparison. Genes with an adjusted $P$-value of ≤ 0.05 were then considered to be differentially expressed for downstream analysis.

## Comparison of RNAseq datasets

To enable a meaningful comparison of datasets, we use a customized ranking factor (RF), defined as:

$$RF = \log_2(\text{Fold} - \text{Change}) \times -\log_{10}(\text{adj.} P - \text{value}).$$

This approach combines both the magnitude of gene expression change (log$_2$ fold change) and the statistical significance (adjusted $P$-value) into a single metric, providing a more robust measure of gene importance. By integrating these two factors, the RF prioritizes genes that exhibit both substantial expression changes and strong statistical support, ensuring that biologically relevant genes are highlighted based on both their expression level and statistical confidence.

## Phosphoproteomics and total proteomics

### Sample preparation for MS analysis

All MS experiments were performed in biological quadruplicates. E14 ES cells were grown on 10-cm dishes and starved of Arg or Leu for 1, 3, or 10 h. A time point 0 h (unstarved) was used as a control. Cells were then washed three times with ice-cold TBS, lysed in 4% sodium deoxycholate (SDC, 30970; Sigma-Aldrich) and 100 mM Tris–HCl, pH 8.5 (T6066; Sigma-Aldrich), and boiled immediately. After sonication, protein amounts were quantified (BCA) and 1.1 mg of each sample was used for digestion. Samples were reduced with 10 mM tris(2-carboxyethyl)phosphine (TCEP), alkylated with 40 mM 2-chloroacetamide (CAA), and digested with trypsin and LysC (1:100, enzyme/protein, w/w) overnight. For proteome measurements, 20 $\mu$g of the total peptide was desalted using SDB-RPS stage tips. 500 ng of desalted peptides was resolubilized in 5 $\mu$l 2% ACN and 0.3% TFA and injected into the mass spectrometer. To enrich phosphorylated peptides, we applied the EasyPhos protocol as previously described (Humphrey et al, 2018). In short, a solution of 50% isopropanol, 6% TFA, and 1 mM monopotassium phosphate (KH2PO4) was added to the digested lysate. Lysates were shaken, then spun down for 3 min at 2,000$g$, and supernatants were incubated with TiO$_2$ beads for 5 min at 40°C (1:10, protein/beads, w/ w). Beads were washed five times with isopropanol and 5% TFA, and phosphopeptides were eluted with 40% acetonitrile (ACN) and 15% of ammonium hydroxide (25% NH$_4$OH) on C8 stage tips. After 20 min of SpeedVac at 45°C, phosphopeptides were desalted on SDB-RPS stage tips and resolubilized in 5 $\mu$l 2% ACN and 0.3% TFA, and injected in the mass spectrometer.

### LC-MS

Samples were loaded onto 50-cm columns packed in-house with C18 1.9 $\mu$M ReproSil particles (Dr. Maisch GmbH), with an EASY-nLC 1000 system (Thermo Fisher Scientific) coupled to the MS (Q Exactive HFX; Thermo Fisher Scientific). A homemade column oven maintained the column temperature at 60°C. Peptides were introduced onto the column with buffer A (0.1% formic acid), and phosphopeptides were eluted with a 70-min gradient starting at 3% buffer B (80% ACN, 0.1% formic acid) followed by a stepwise increase to 19% in 40 min, 41% in 20 min, 90% in 5 min, and 95% in 5 min, at a flow rate of 300 nl/min, whereas peptides for proteome analysis were eluted with a 120-min gradient starting at 5% buffer B (80% ACN, 0.1% formic acid) followed by a stepwise increase to 30% in 95 min, 60% in 5 min, 95% in 2 × 5 min, and 5% in 2 × 5 min at a flow rate of 300 nl/min. A data-independent acquisition MS method was used for proteome and phosphoproteome analysis in which one full scan (300–1,650 $m/z$, R = 60,000 at 200 $m/z$) at a target of 3 × $10^6$ ions was first performed, followed by 32 windows with a resolution of 30,000 where precursor ions were fragmented with higher energy collisional dissociation (stepped collision energy 25%, 27.5%, 30%) and analyzed with an AGC target of 3 × $10^6$ ions and a maximum injection time at 54 ms in profile mode using positive polarity.

### MS data analysis

MS raw files were processed by Spectronaut software version 13 (Biognosys) using the mouse UniProt FASTA database (22,220 entries, 39,693 entries, 2015). Proteome and phosphoproteome files were analyzed via direct DIA. For proteome analysis, standard settings were used. The false discovery rate (FDR) was set to less than 1% at the peptide and protein levels, and a minimum length of 7 AAs for peptides was specified. Enzyme specificity was set as C-terminal to Arg and L-Lys as expected using trypsin and LysC as proteases and a maximum of two missed cleavages. For the phosphoproteome analysis, serine/threonine/tyrosine phosphorylation was added as variable modification to the default settings, which include cysteine carbamidomethylation as fixed modification and N-terminal acetylation and methionine oxidations as variable modifications. The localization cutoff was set to 0.

### Bioinformatics data analysis

We performed downstream analyses in the Perseus software version 1.6.2.2 (Tyanova et al, 2016). For phosphosite analysis, Spectronaut normal report output tables were collapsed to phosphosites and the localization cutoff was set to 0.75 using the peptide collapse plug-in tool for Perseus as previously described (Bekker-Jensen et al, 2020), which collapses phosphoions to phosphosites. Importantly, it does not sum up the intensities of a phosphosite on peptides, if different phosphorylations are also present. For example, the intensity of Larp1_S743_M1 and Larp1_S743_M2 differs because although Larp1_S743_M1 represents the singly phosphorylated peptide, Larp1_S743_M2 reflects two phosphorylated sites on one (or more) peptides containing S743 (Larp1_S743_S751 and Larp1_S743_T747 in this case). For each phosphosite on a multiple phosphorylated peptide, we receive a row with the same intensities as these phosphorylations are localized on the

same peptide. Eif4ebp1_S64_M2 and Eif4ebp1_T69_M2 share the same intensity as they represent the two phosphosites on the same peptide. Different phosphosites on the same peptide can have slightly different fold changes because of imputation. Also, each collapse key (gene_position_multiplicity) is unique, which means that if a phosphosite is present on two peptides that carry a different phosphosite, just one row will be assigned. Phosphosites located on phosphopeptides with more than two phosphorylations are labeled with a multiplicity of three (M3). Summed intensities were $log_2$-transformed. Quantified proteins and phosphosites were filtered for at least 75% of valid values among three or four biological replicates in at least one condition. Missing values were imputed, and significantly up- or down-regulated hits were determined by multiple-sample test (one-way ANOVA, $s0$ = 0.1, FDR = 0.05) and $t$ test (two-sided, $s0$ = 0.1, FDR = 0.05). For hierarchical clustering of significant proteins and phosphosites, median abundance of biological replicates was z-scored and clustered using Euclidean as a distance measure for row clustering. Fisher's exact tests were performed to detect the systematic enrichment of annotations and pathways by analyzing proteins whose levels or expression or phosphorylation levels are significantly regulated upon different conditions. We used KEGG names, motifs, and PhosphoSitePlus (PSP) kinase/substrate annotations, and either the $P$-value or the Benjamini–Hochberg FDR was set to 0.02 as a threshold. The PSP database contains 2,171 known mouse kinase/substrate annotations (including 87 mTOR kinases/substrates). The identity and position of the phosphorylation sites on 4ebp1 and Ulk1 were inferred from PSP.

## Seahorse bioenergetic measurements

25,000 ES cells were plated into each well of Seahorse cell culture plates precoated with 0.1% gelatin. 5 h later, starvation experiments were started as described above and carried out overnight (17 h). Real-time oxygen consumption rate measurements were made with Seahorse XF HS Mini Analyzer (Agilent) and the Seahorse XFp Cell Mito Stress kit (103010-100; Agilent) following the manufacturer's instructions. Briefly, before running the assay, the plates were preincubated at 37°C for a minimum of 45 min in the absence of $CO_2$ in Seahorse XF DMEM (103575-100; Agilent) supplemented with 2 mM glutamine, 10 mM glucose, and 1 mM pyruvate. Cells were stimulated with 1.5 $\mu$M oligomycin, 1 $\mu$M carbonyl cyanide 4-(trifluoromethoxy)phenylhydrazone (FCCP), and 0.5 $\mu$M antimycin A/rotenone. After the experiment, cells were lysed in RIPA and the protein amounts quantified in each sample to allow normalization between conditions.

## Targeted metabolomics

### Sample preparation

All samples were prepared in biological triplicates. ES cells were starved of Arg or Leu for 1, 3, and 10 h as described before, and medium containing all amino acids was used as a control. After the indicated time points, cells were washed with PBS and lysed using a precooled MeOH:ACN:$H_2O$ (2:2:1, vol/vol) solvent mixture (MeOH and ACN were ACS grade and from Sigma-Aldrich and AppliChem,

respectively). Cell lysates were then vortexed for 30 s and incubated in liquid nitrogen for 1 min. The samples were then allowed to thaw at room temperature and sonicated for 10 min in a water bath sonicator. The cycle of cell lysis in liquid nitrogen combined with vortex and sonication was repeated two times. To precipitate proteins, the samples were incubated for 1 h at –20°C, followed by a 15-min centrifugation at 9,000$g$ at 4°C. After centrifugation, the supernatants were snap-frozen in liquid nitrogen and stored at –80°C.

### LC-MS

Metabolite extracts have been analyzed by either reversed-phase chromatography or hydrophilic interaction chromatography (HILIC), both directly coupled to mass spectrometry (LC-MS/MS). For reversed-phase LC–MS/MS, 100 μl of the extracts has been dried down in a vacuum centrifuge and resolved in 100 μl of 0.1% formic acid in water. For each sample, 1 μl was injected onto a Kinetex (Phenomenex) C18 column (100 Å, 150 × 2.1 mm) connected with the respective guard column and employing a 7-min-long linear gradient from 99% A (1% acetonitrile, 0.1% formic acid in water) to 60% B (0.1% formic acid in acetonitrile) at a flow rate of 80 μl/min. Detection and quantification has been done by LC-MS/MS, employing the selected reaction monitoring (SRM) mode of a TSQ Altis mass spectrometer (Thermo Fisher Scientific), using the following transitions in the positive ion mode: 76 to 30 m/z (glycine), 90 to 44 m/z (alanine), 106 to 60 m/z (serine), 116 to 70 m/z (proline), 118 to 72 m/z (valine), 120 to 74 m/z (threonine), 122 to 76 m/z (cysteine), 132 to 86 m/z (leucine and isoleucine),133 to 74 m/z (asparagine), 134 to 74 m/z (aspartic acid), 147 to 84 m/z (lysine), 147 to 130 m/z (glutamine), 148 to 84 m/z (glutamic acid), 150 to 133 m/z (methionine), 156 to 110 m/z (histidine), 166 to 133 m/z (phenylalanine), 175 to 70 m/z (arginine), 182 to 36 m/z (tyrosine), 205 to 188 m/z (tryptophan), 241 to 74 m/z (cystine). In HILIC, 1 μl of the original sample was directly injected onto a polymeric iHILIC-(P) Classic HPLC column (HILICON, 100 × 2.1 mm; 5 μm) and the respective guard column, operated at a flow rate of 100 μl/min. The HPLC (Ultimate 3000 HPLC system; Dionex; Thermo Fisher Scientific) was directly coupled via electrospray ionization to a TSQ Quantiva mass spectrometer (Thermo Fisher Scientific). A linear gradient (A: 95% acetonitrile 5% 10 mM aqueous ammonium acetate; B: 5 mM aqueous ammonium bicarbonate) starting with 15% B and ramping up to 60% B in 9 min was used for separation. The following SRM transitions were used for quantitation in the negative ion mode: 87 to 43 m/z (pyruvate), 140 to 79 m/z (carbamoyl phosphate), 191 to 111 m/z (citrate), 229 to 97 m/z (pentose phosphates), 259 to 97 m/z (hexose phosphates), 426 to 134 m/z (ADP), 506 to 159 m/z (ATP), 662 to 550 m/z (NAD), 664 to 408 m/z (NADH), 742 to 620 m/z (NADP), 744 to 426 m/z (NADPH), 808 to 408 m/z (acetyl-CoA).

### Data analysis

Data interpretation was performed using TraceFinder (Thermo Fisher Scientific). Authentic standards were used for determining collision energies and for validating experimental retention times via standard addition. The metabolite peaks from raw HPLC-MS chromatograms were integrated in the software by evaluating the extracted ion chromatogram (XIC) counts. All datasets were normalized to the amount of protein in the samples (determined by the Pierce BCA protein assay kit, 23227; Thermo Fisher Scientific) to account for cell growth. Data are shown as XIC area/μg of protein, $\log_2$-transformed result as indicated in the figure legends.

### Statistics and reproducibility

The statistical analyses shown in bar plots were performed with GraphPad Prism 9 using one-way or two-way ANOVA followed by Dunnett's test for multiple comparisons, unless otherwise mentioned. A $P$-value less than 0.05 was considered significant. All non-omics experiments were independently repeated more than two times with similar results. Omics-related statistical analyses are described in the previous sections above.

## Data Availability

All the MS data generated for this study have been deposited to the ProteomeXchange Consortium via the PRIDE repository (http://www.proteomexchange.org/) with the identifier PXD030959. RNA-seq data have been deposited in the Gene Expression Omnibus repository (https://www.ncbi.nlm.nih.gov/geo/) under the accession number GSE291532. All the remaining data are available within the source data or from the corresponding author upon reasonable request. The MPIB NGS Core Facility (RRID:SCR_025746), MPIB Mass Spectrometry Facility (RRID:SCR_025745) and MPIB Bioinformatics Core Facility (RRID: SCR_025742) were used for this project.

## Supplementary Information

## Acknowledgements

We thank Professors Brenda Schulman and Danny Nedialkova at MPIB for reagents, and Sabine Suppmann and the MPIB Protein Production Core Facility for LIF. We also acknowldege Dr. Rin Ho Kim and the MPIB NGS Core Facility for RNAseq sample preparation, data acquisition and processing and Dr. Barbara Steigenberger of the MPIB Mass Spectrometry Core Facility. We thank Dr. Thomas Köcher and the Vienna BioCenter Core Facilities (VBCF, Vienna, Austria) for metabolomics sample preparation, and data acquisition and processing. The VBCF Metabolomics Facility is funded by the City of Vienna through the Vienna Business Agency. We thank Hiromune Eto for helping with RNAseq analysis and Leo Kiss, Leonie Zeitler, Johanna Brüggenthies, and Patricia Ogger for scientific input. Work in the Murray laboratory is supported by the Max-Planck-Gesellschaft.

### Author Contributions

M Russier: conceptualization, data curation, formal analysis, investigation, methodology, and writing—review and editing.
A Fiore: formal analysis, investigation, methodology, and writing—review and editing.

A Bici: validation, investigation, and writing—review and editing.

A Groß: investigation and methodology.

M Tanzer: data curation, formal analysis, validation, investigation, and methodology.

A Yeroslaviz: data curation, formal analysis, and methodology.

M Mann: resources, supervision, and methodology.

PJ Murray: conceptualization, resources, data curation, formal analysis, supervision, funding acquisition, validation, investigation, methodology, project administration, and writing–original draft, review, and editing.

## Conflict of Interest Statement

The authors declare that they have no conflict of interest.

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
