## [Reviewer comments · Life Science Alliance]

Life Science Alliance

Differential cell survival outcomes in response to diverse amino acid stress

Marion Russier, Alessandra Fiore, Ana Bici, Annette Groß, Maria Tanzer, Assa Yeroslaviz, Matthias Mann, and Peter Murray
DOI: <https://doi.org/10.26508/lsa.202503324>

Corresponding author(s): Peter Murray, Max Planck Institute of Biochemistry

Review Timeline:

Submission Date:	2025-03-25
Editorial Decision:	2025-05-19
Revision Received:	2025-07-25
Editorial Decision:	2025-08-22
Revision Received:	2025-08-25
Accepted:	2025-08-25

Scientific Editor: Tim Fessenden

Transaction Report:

May 19, 2025

Re: Life Science Alliance manuscript #LSA-2025-03324

Prof. Peter Murray
Max Planck Institute of Biochemistry
Immunoregulation Research Group
Am Klopferspitz 18
Martinsried 82152
Germany

Dear Dr. Murray,

Thank you for submitting your manuscript entitled "Differential cell survival outcomes in response to diverse amino acid stress" to Life Science Alliance. The manuscript was assessed by expert reviewers, whose comments are appended to this letter. We invite you to submit a revised manuscript addressing the Reviewer comments noted below.

As you will see, all reviewers appreciated the important new results in amino acid sensing shown here. Reviewers 2 and 3 made minor requests on data interpretation and figure presentation which we invite you to consider towards improving this work. Reviewer 1 raised several concerns on specific claims made and on data availability and interpretation. In particular, this reviewer asked if the reported changes in metabolites are due to altered production or consumption (point 3b), and requested testing if GCN2 is involved in cell death following Cys withdrawal (point 4). This reviewer and Reviewer 2 observed that claims made from observations with torin should be tempered given potential off-target effects of this compound. Finally, all reviewers remarked, in different forms, that the evidence and novelty needed to name a completely novel form of cell death are not in place in this work. While a revised manuscript should respond to all comments in some form, additional data beyond those noted here are left to your discretion.

While you are revising your manuscript, please also attend to the below editorial points to help expedite the publication of your manuscript. Please direct any editorial questions to the journal office. The typical timeframe for revisions is three months. Please note that papers are generally considered through only one revision cycle, so strong support from the referees on the revised version is needed for acceptance.

Thank you for this interesting contribution to Life Science Alliance. We are looking forward to receiving your revised manuscript.

Sincerely,

B. MANUSCRIPT ORGANIZATION AND FORMATTING:

Reviewer #1 (Comments to the Authors (Required)):

The manuscript "Differential cell survival outcomes in response to diverse amino acid stress" by Russier et al. provides insights into amino acid (AA) deprivation in non-cancerous pluripotent cells. The findings support the idea that deprivation of specific AA elicits unique adaptation mechanisms. The authors studied the response to arginine (Arg) and leucine (Leu) withdrawal in murine ES cells at transcriptome, proteome, and phosphoproteome level. Upon Leu or Arg starvation, the cells exit the proliferative state, leading to quiescence and metabolic adaptation. The quiescent state is reversible upon readdition of the AA. The authors found that several transcripts, proteins, and phosphosites changed under starvation. Many upregulated proteins were involved in lysosome biogenesis, and proteins relating to ribosome biogenesis were downregulated. The authors evaluated the contribution of GCN2 to the transcriptomic changes by GCN2 KO and RNA seq analysis. They found that the expression of known integrated stress response proteins like Txnip1 and Ddit3 depended on GCN2. However, transcripts of lysosomal and metabolic enzymes were induced independently of GCN2. The authors note that while Arg and Leu are known to have different upstream regulators, the responses to starvation at various omic levels were comparable. This similar response contrasts with observations in transformed cells where Arg withdrawal produces a more robust transcriptional response than Leu. The authors also investigated Cysteine (Cys) and Threonine (Thr) starvation and found that both conditions elicit cell death. Cys starvation leads to ferroptosis, whereas Thr deprivation leads to a new type of cell death, which can be suppressed by Met depletion, mTOR, or translation inhibition.

This study presents interesting data, however many of the statements and claims are not supported by evidence and need to be revised. This is particularly important as the authors claim ES cells to be different from transformed cells, and hence it is not sufficient to substantiate claims based on literature evidence from transformed cells. This is for instance evident for the claim of a new form of cell death, termed "threoptosis": as detailed below, the evidence for this type of cell death as well as for the novelty of this finding is not sufficient

There are several conceptual and technical issues:

Conceptual issues

(1) The authors claim that ES cells are a suitable model to test the effects of AA withdrawal as they are non-transformed, non-cancerous cells. Murine ES cells heavily rely on Thr as an energy source via the TDH enzyme. This enzyme seems non-functional in primates. The authors should discuss whether this is a limitation of using murine ES cells and if the results would be consistent in human cells.

(2) The authors propose threoptosis as a novel cell death pathway, but it is known since over a decade that Thr catabolism elicits a specialized type of cell death in murine ES cells (Alexander et al, PNAS, 2011, PMID 21896756). The authors should cite this literature and analyze whether they observe the same type of cell death, which seems likely. They should also investigate whether this type of cell death is specific for murine ES cells or if it also occurs in other non-transformed or transformed cell types.

(3) The omics datasets are presented in a descriptive manner, and the coherence is missing:

(a) It would be advisable to use multiomic data analysis tools (e.g. MOFA, mixOmics) to identify which layer contributes significantly to AA withdrawal. Is it the expression of the proteins, or the changes in phosphoproteome?

(b) It is unclear in the targeted metabolomic analysis if the decrease in the metabolites is because their production was lower or if they were rapidly consumed. The authors should discuss this issue and conduct pathway analyses to investigate whether the effects on metabolic endpoints which they observe can be more likely assigned to catabolic or anabolic processes.

(c) Some of the isoenzymes, like Hk1 and Adpk1, are differentially regulated. The authors should elaborate on the significance of these changes, and on the impact and connection to metabolic profiles.

(d) The authors state that "the transcriptomes and proteomes were largely uncorrelated (Fig. S2a, b)", and they assign this to reduced bulk translation under AA starvation. What is the relevance of transcriptome changes if they do not translate to the proteome? Which conclusions are based on the transcriptome analyses, and can they be validated at protein level? If so, do the conclusions change based on the transcriptome data or is it sufficient to investigate the proteome and omit the transcriptome under AA deprivation? These questions should be analysed by multi-omic data integration, and the results should be presented and discussed.

(4) The authors claim that Cys starvation-mediated cell death is prevented by translation inhibitors, showing that this effect is translation dependent. The authors should test if the effects are sensitive to GCN2-mediated translation. They can conduct this analysis in the GCN2 KO cells which they have already.

(5) The authors observe that Met withdrawal protects the ES cells from Cys/Thr starvation induced cell death, and they show similar effects for translation inhibitors. They conclude that Met withdrawal protects cells by translation inhibition, however they do not provide evidence supporting this causal link. The authors should analyse their datasets regarding effects on the translation machinery under combined Cys/Thr+Met deprivation and provide mechanistic data to support their claim.

(6) The authors mention that Leu and Arg deprivation inhibits mTORC1 via activation of Sestrin-2 and Castor-2, but there is no data-based follow up on this. Please mark Sestrin-2 and Castor-2 in the proteome and phosphoproteome data. Please also include detections of Sestrin-2 and Castor-2, and data on mTORC1 activity (substrate phosphorylations) for the different time points of starvation.

Technical issues and FAIR data

(1) The authors should upload the uniprot fasta file used for the proteome and phosphoproteomic analysis to PRIDE. Further, an SDRF file containing information on which file belongs to which condition should be added.

(2) Several controls are missing and data are often overinterpreted. Issues are discussed here based on the example of Fig. 4a, 1b and S1c. The issues should be solved throughout the manuscript:

a. In 4a:

- the authors claim based on 4E-BP1-T37/46 phosphorylation that rapamycin does not inhibit mTORC1 and translation, whereas the ATP analogue mTORC1 inhibitor Torin-1 does. The effect shown on 4E-BP1-pT37/46 is very well described and it is known to be mediated by the specific mode of 4E-BP1 binding to mTORC1 (PMID: 33852892). The data does not allow to conclude that rapamycin does not inhibit translation; nor does the data with Torin-1 allow to conclude that the Torin-1 effect is mTORC1 or translation mediated, as Torin-1 inhibits both mTOR complexes, and it has off target effects e.g. on PI3 kinases. The authors should correctly present literature knowledge and avoid overinterpretation of their data.
- Other mTORC1 readouts including S6K-pT389 should be detected throughout to support claims on mTORC1 activity.
- Furthermore, quantification and statistics as well as indication of the number of replicates is missing and need to be added.
- Also, the total level is missing for 4E-BP1, and total levels should be detected throughout for all phosphorylated proteins.
- The loading control is missing.
- The significance of TSC2 detection in this figure is unclear as it is not adequately discussed. Either it should be discussed or left out.

b. Fig. 1b: The puromycin assay shows that protein synthesis is reduced upon Leu or Arg starvation, yet there is a difference between Arg and Leu starvation that is not discussed. Please provide quantification, statistics and number of replicates. Furthermore, Grb2 obviously is not an appropriate loading control as it is downregulated upon AA starvation. Other proteins, such as Tubulin, should be tested for their suitability as loading controls.

c. Fig. S1c: GCN2 and ATF4 are known to act in the integrated stress response. Accordingly, GCN2 is phosphorylated and ATF4 levels are enhanced after 3 h of the different types of AA starvation. However, ATF4 levels reach the peak after 3 h of starvation and then decline and in the case of Leu starvation, ATF4 vanishes after 24 h. The divergence should be discussed

and detections of phospho-GCN2 should be shown for longer time points.

(3) Legends

- Fig. S3c. What does the legend refer to? There are 3 graphs but only one legend. Please revise.
- Fig. 3b. The figure seems to refer to 3a, yet the legends don't match. Please revise.

(4) Terminology: Line 7: "protein translation" - proteins are not being translated. The correct term is either mRNA translation or protein biosynthesis.

Reviewer #2 (Comments to the Authors (Required)):

In this study, Russier et al. investigate the cellular response to removal of individual amino acids (AA). Using a comprehensive multi-omics approach, they describe how mouse ES cells (mESCs) respond to deprivation of Arg or Leu, by entering a quiescence-like mode without dying. Interestingly, they show that the response to Cys or Thr removal is distinct as cells deprived of these AA for longer times undergo cell death, mainly due to an imbalance between low nutrient supply and anabolic protein synthesis.

This is an interesting study, addressing an important and timely topic in the AA sensing field, which is how cells adapt to nutritional changes in their environment. In particular, the differential responses to distinct AA are intriguing and nicely complement recent work on compartmentalized mTOR signaling in cells. Specific suggestions to improve this manuscript further before publication are listed below.

Major comments

- Fig. 1a-c: The authors observe no strong differences in the response of cells to Leu or Arg deprivation. Given that Leu is an essential amino acid, while Arg is not (ie it can be synthesized by cells through metabolic adaptations), how do the authors explain these effects? What happens to pathways involved in arginine metabolism in Arg-deprived cells? The authors should at least discuss this further in the text.
- Page 9, lines 22-26, the authors mention that Torin1 triggers translation inhibition similar to that caused by Arg or Leu starvation. Moreover, starvation of either AA is sensed via mTORC1, which is inhibited by Torin1. Yet, Arg or Leu starvation fail to prevent 'threoptosis', while CHX treatment or Torin1 treatment is capable of blocking it. Is the proteome differently rewired between these nutritional and pharmacological strategies? How do the authors explain this discrepancy?
- A recent study described the separation of lysosomal and cytoplasmic mTORC1 functions towards distinct substrates. In line with the findings described here, Leu/Arg seem to primarily control lysosomal mTORC1, whereas Thr/Cys signal to non-lysosomal mTORC1 complexes (PMID: 39385049). How is starvation of the different AA expected to affect phosphorylation of lysosomal and non-lysosomal mTORC1 targets in this study? Minimally, the authors should discuss how their findings fit with previous work on AA sensing.
- Fig 1a: Can the authors explain why Ki67 intensity increases between 1h and 8h in the +AA samples? Does this mean that the main difference between the two time points is the changes under AA-replete conditions and not the response to Arg or Leu removal?
- Fig 1c: Why does AA re-addition at 10 and 24 hours boost cell confluence to a higher extent compared to non-starved cells (0 h time point of re-addition)?
- Fig. 4: The identification of a potentially novel mode of cell death ('Threoptosis') is intriguing. However, molecular players that are involved in this process are not investigated in this study. As blockage of translation prevents threoptosis, this could include translation initiation or elongation factors, mTOR signaling components, or ribosomal proteins. Although this is likely beyond the scope of this work, what are the proteomic and/or metabolic changes that occur if one compares 0h to 8h of Thr deprivation (ie before cell death is reported)? Such an approach would likely identify factors specifically related to 'threoptosis'.
- Fig. 4: Removal of Met demonstrates a protective effect in Thr-starved cells, preventing the activation of 'threoptosis'. How can the authors exclude that this effect is not due to mTORC1 inhibition by Met removal, eg via the previously described SAMTOR-related pathway?
- Fig 4a-b: Given the effect of Torin1 but not Rapamycin in cell proliferation, as well as its known role in cell survival, can the authors exclude the involvement of mTORC2 in 'threoptosis'? Or is this due to the difference between rapamycin and torin1 in the regulation of rapamycin-resistant substrates like 4E-BP1? Please discuss this further in the text.

- The authors fail to cite and discuss a relevant recent study that described the effects of Leu and Arg availability in mESC proliferation, cell cycle arrest and pluripotency (PMID: 36430764). How do these data fit with the work presented here?

Minor comments

- Fig S3c: The annotation of ERK1/2 and mTOR/S6K substrates is missing from the plots (all data points are gray).
- Fig 1f: It would make the figure more reader-friendly if the different data point groups were also labelled directly in the plots.
- Fig S5d: The x-axis in the lower Leu-Cys plot is mislocalized at 100 instead of 0.
- Fig S6c: The figure legend does not seem to correspond to the data shown in the panel.
- Fig 4g is wrongly referred to as 4h in the text.

Reviewer #3 (Comments to the Authors (Required)):

The article "Differential cell survival outcomes in response to diverse AA stress" by Russier et al. provides a thorough analysis of the response of embryonic stem cells to amino acid starvation. Specifically, they used a combination of proteomic, metabolomic, and transcriptomic methods to determine and compare the effects of limiting the individual amino acids Arginine, Leucine, Threonine, and Cysteine. This is an elegant, well-designed, and timely report. Among several interesting contributions, this work provides a comprehensive picture of amino acid starvation in fast proliferating but non-transformed, providing an important counterpoint to our knowledge of amino acid starvation in malignant cells. The difference in response to different amino acids - and specially the response to threonine deprivation are quite interesting. As the authors acknowledge in their discussion, it would be ideal to study the effects of limiting additional amino acids and their combinations. While attractive, a more exhaustive analysis of amino acid starvation will quickly scale up to unfeasibility. Still, this current work provides an intriguing opening to the mechanisms and strategies cells may use to cope with the deprivation of different amino acids. I think this is an article well worth to be published so below I only provide minor notes hoping that they may improve the manuscript.

- The more detailed analysis of the biological consequences and mechanisms behind the responses to different amino acids are beyond the scope of this work. Still, the authors show very interesting results with the dual Threonine/Methionine removal. I wonder if they would observe a similar death (with Thr) and rescue (with Met) in cells KO for THD. Investigating the role for this enzyme in cell death triggered by Thr may be relevant for human cells - that as they mention - do not express this enzyme.
- I am not a big fan of finding a new name for cell death processes. But that is more a personal preference. While the authors present evidence to rule out mechanisms such as apoptosis and necroptosis, there are already many other forms of cell death. Is it possible that cells fail to sense the lack of Thr and thus they continue translation as normal which then causes lethal ER stress? That would be consistent with the effects of Torin and the reduction of translation.
- It would be nice to have some additional cross-talk between their omics and more mechanistic studies. For example, do they detect an increase in GCN2 phosphorylation during -Arg and -Leu in their phosphoproteomic analysis?
- A small detail but it may be better to mention that Arg is a conditional essential amino acid? Also, it may be worth mentioning that ES cells have special amino acid needs/mechanisms including the expression of glutamine synthase and thus are able to survive without Gln supplementation.
- Are columns in Fig. 1h/I also ordered by hierarchical clustering? It is fine either way but to me it is not clear if the column order emerges from the data, or the columns are manually organized by treatment/temporal response. Again, I think either would be OK.
- In Figure 1g, it is unclear what the gray boxes represent (maybe the number of proteins/phosphoproteins/transcripts that were not significantly different?). I'd suggest clarifying this. In the same figure, in the left plot, there is a lost number (2041) below the blue box in the middle. I'm suspecting it belongs in the second panel, on top of the blue box in the middle.
- In the GCN2 blots shown in S1 it would be better to use total GCN2 to highlight the increase in phospho GCN2.
- Figure S3c: there is a legend for phospho-sites from Erk1/2 and mTOR/p70s6kb substrates, but no colored dots are visible in the plot (that I can see at least).

Overview of the Rebuttal:

We appreciate the time and effort each reviewer spent assessing our manuscript. All the reviewers noted many positive elements of our study. In fact, Reviewers 2 and 3 raised issues that can be mainly addressed by changes in the text, or clarification of figures and text. Some comments from the reviewers requested broad speculation about our experimental system and the outcomes we report here, which we take as a sign of interest in our work. However, one manuscript cannot answer all the fundamental questions about a complex biological process. We assume that the requests for speculation (or vastly complex experimentation) are indications of the value of understanding a cell biological system of stress regulation.

A point of contention was our use of the term threoptosis to indicate a specific form of cell death. In response, we have removed the term “threoptosis” and instead used the term “threonine deprivation-associated cell death”, which is not as succinct but avoids the issue of trying to assign a new death process. We hope the reviewers agree that this more conservative approach is reasonable.

Reviewer 1 asked for an experiment in the GCN2 KO ES cells that we also performed using GCN2-IN-6 and this is discussed below. Reviewer 1 also raised a number of points which require extensive speculation or were already discussed in the manuscript; we address these comments in sequence below. We would prefer to refrain from expanding the scope and length of the manuscript too much.

Reviewer 2 suggested we explore the role of the mTOR complexes further. We have performed experiments with different inhibitors and these data are reported in the manuscript (Fig. S6d)

Specific responses to the Reviewer comments are shown under each comment in blue.

Reviewer #1

(1) The authors claim that ES cells are a suitable model to test the effects of AA withdrawal as they are non-transformed, non-cancerous cells. Murine ES cells heavily rely on Thr as an energy source via the TDH enzyme. This enzyme seems non-functional in primates. The authors should discuss whether this is a limitation of using murine ES cells and if the results would be consistent in human cells.

The reviewer asks for a discussion of the species-specific use of THD. However, others have extensively covered this issue. We cited the original, and key, reference from the McKnight laboratory (now ref. 44). Since the mouse-specific (in mammals) use of TDH in ES cells is well-known, we consider that the citation of the original finding and a brief note is sufficient.

(2) The authors propose threoptosis as a novel cell death pathway, but it is known since over a decade that Thr catabolism elicits a specialized type of cell death in murine ES cells (Alexander et al, PNAS, 2011, PMID 21896756). The authors should cite this literature and analyze whether they observe the same type of cell death, which seems likely. They should also investigate whether this type of cell death is specific for murine ES cells or if it also occurs in other non-transformed or transformed cell types.

As noted above, we are well aware of the literature on THD. The PNAS paper the reviewer notes is a follow up paper to the original paper (ref 44) in *Science* that we cite. The reviewer asks for whether this cell death occurs in other cells. Our manuscript is about ES cells. Expansion of our findings to other cell types and species is a valid request and potentially important (e.g., Trypanosomes use THD to metabolize threonine) but beyond the scope of our focused work.

(3) The omics datasets are presented in a descriptive manner, and the coherence is missing:

(a) It would be advisable to use multiomic data analysis tools (e.g. MOFA, mixOmics) to identify which layer contributes significantly to AA withdrawal. Is it the expression of the proteins, or the changes in phosphoproteome?

We appreciate the suggestion to apply integrative multi-omics tools, which could indeed provide further insight into which molecular layer, protein expression or phosphoproteome changes, plays a more relevant role in response to AA withdrawal. While this specific analysis was beyond the scope of the current study, we have made the original datasets publicly available, and we welcome further exploration using these or other emerging tools developed by the bioinformatics community.

(b) It is unclear in the targeted metabolomic analysis if the decrease in the metabolites is because their production was lower or if they were rapidly consumed. The authors should discuss this issue and conduct pathway analyses to investigate whether the effects on metabolic endpoints which they observe can be more likely assigned to catabolic or anabolic processes.

This important question is best answered by targeted metabolomics for arginine and leucine of the cell lysates and supernatants across time (part of the experimental series in Fig. 1d). As shown in Rebuttal Figure 1 (top 4 graphs), the absence of arginine or leucine causes complete depletion of detectable amino acid across time. In other words, when ES cells are incubated with -Leu media, the amount of Leu detected in the media or in the cells is below the limit of detection (same is true for -Arg media).

Our interpretation of this data is straightforward: (i) the experimental system proves the complete depletion of each amino acid, validating the approach. (ii) Given that arginine or leucine can conceivably be recovered from different sources (as detailed in the introduction), their use in residual translation of other process is either rapid (i.e., the flux is sufficiently rapid that residual amino acid is undetectable) or non-existent. The reviewer asks about “fates” (e.g., ornithine from arginine). However, we consider that in the absence of detailed isotope/isotopologue tracing for arginine or leucine, any interpretation is speculative and will not support the messages of the manuscript.

There is an interesting aspect of this data that we cannot yet explain, which is why we omitted the data from the manuscript – namely, when leucine is removed, the amount of arginine detected increases. The same is true in reverse (e.g., see the detection of leucine in the top left graph when arginine is removed). A simple hypothesis to account for this observation is that once one amino acid is removed, autophagy and proteolysis recover pools of amino acids – these are only used slowly because translation has ceased. However, and to reiterate from above, a considerable amount of work is needed to track the reservoirs and fates of each amino acid.

(c) Some of the isoenzymes, like Hk1 and Adpk1, are differentially regulated. The authors should elaborate on the significance of these changes, and on the impact and connection to metabolic profiles.

The reviewer requests speculation on the significance of changes on enzymes in Fig. 2g. However, without extensive experimentation, we prefer to avoid speculation on individual transcripts/proteins that are altered in our comprehensive datasets that would require detailed and individualized experimentation.

(d) The authors state that "the transcriptomes and proteomes were largely uncorrelated (Fig. S2a, b)", and they assign this to reduced bulk translation under AA starvation. What is the relevance of transcriptome changes if they do not translate to the proteome? Which conclusions are based on the transcriptome analyses, and can they be validated at protein level? If so, do the conclusions change based on the transcriptome data or is it sufficient to investigate the proteome and omit the transcriptome under AA deprivation? These questions should be analysed by multi-omic data integration, and the results should be presented and discussed.

The reviewer asks a question that has bedeviled integrated stress response (ISR) researchers for ~30 years. The key questions here concern (1) which transcripts induced by the ISR are actually translated? (2) Of the translated proteins (under amino acid stress), what are their functions? What is their stability? (3) Why are so many transcripts induced during a period of overall reduced translation? To answer these questions, an integrated approach to measure (i) de novo mRNA production during amino acid stress across time, (ii) de novo protein synthesis during amino acid stress across time, (iii) the total mRNA and total protein amounts across time and (iv) a genetic strategy to find the key genes/proteins necessary for these changes. Ideally, such an experimental strategy would use multiple human and mouse cell systems (transformed and primary, with multiple conditions and timepoints). Further single cell approaches would be valuable to discern heterogeneity between cells. In our view, the scope and size of such an approach is vast and exceeds the ability of a small and focused laboratory. Thus, "these questions should be analysed by multi-omic data integration" could be viewed as a request to expand the scale and scope of our work far beyond the original intent and focus.

(4) The authors claim that Cys starvation-mediated cell death is prevented by translation inhibitors, showing that this effect is translation dependent. The authors should test if the effects are sensitive to GCN2-mediated translation. They can conduct this analysis in the GCN2 KO cells which they have already.

We performed experiments with Cys starvation in WT vs. GCN2 knockout ES cells as requested (Rebuttal Fig. 2). In addition, we used GCN2-IN-6 in the WT cells as an addition verification of the potential role of GCN2. The results are shown in rebuttal Figure 2 as representative data from four independent experiments. The results show that GCN2 may have a minor effect on suppressing -Cys-induced ferroptosis. However, the absence of GCN2 function delays the onset of cell death by 1-2 hr and thereafter the cells die with similar kinetics to the WT controls for either GCN2 KO cells or cells treated with GCN2-IN-6. Based on these findings, we consider the Reviewer's question has been addressed. However, these data are not sufficiently compelling to add to the manuscript.

(5) The authors observe that Met withdrawal protects the ES cells from Cys/Thr starvation induced cell death, and they show similar effects for translation inhibitors. They conclude that Met withdrawal protects cells by translation inhibition, however they do not provide evidence supporting this causal link. The authors should analyse their datasets regarding effects on the translation machinery under combined Cys/Thr+Met deprivation and provide mechanistic data to support their claim.

We do not understand this request. “effects on the translation machinery” is not clear. Our results show a clear effect of Met starvation. However, finding the root cause of this effect may require extensive experimentation and is beyond the scope of a concise report.

(6) The authors mention that Leu and Arg deprivation inhibits mTORC1 via activation of Sestrin-2 and Castor-2, but there is no data-based follow up on this. Please mark Sestrin-2 and Castor-2 in the proteome and phosphoproteome data. Please also include detections of Sestrin-2 and Castor-2, and data on mTORC1 activity (substrate phosphorylations) for the different time points of starvation.

In the manuscript, we made an error with naming – Castor-1 is the arginine binding sensor discovered by the Sabatini group; this has now been corrected). As shown in Rebuttal Figure 3, we analyzed proteome data across time during leu and arg deprivation, where each time point is compared to the untreated, unstarved control. Notably, Castor-1, Sesn2, SAMTOR, and GCN2 were not detected in the total proteome. These proteins were also not identified in the phosphoproteome data (not shown). Although Sesn2 mRNA is induced as part of the ISR transcriptional response, this induction does not appear to translate into detectable protein under these conditions. As an aside, we have been working on both proteins for some time and generated ES cells lacking Sesn2 or Castor1. The absence of either “sensor” has a minor effect of mTORC1 signaling via leucine or arginine in ES cells and the absence of either protein has no effect on the “quiescence” response of ES cells to leucine or arginine deprivation. This data, and a related series of many experiments, thus did not provide conclusive or compelling information on the connections between the upstream mTORC1 amino acid sensors, mTORC1 or the downstream response of the ES cells to starvation. Accordingly, the data shown here, and related experiments were omitted. This information is provided for the reviewers only.

Technical issues and FAIR data

(1) The authors should upload the uniprot fasta file used for the proteome and phosphoproteomic analysis to PRIDE. Further, an SDRF file containing information on which file belongs to which condition should be added.

Our proteomic data was already uploaded to PRIDE. However, later, we realized that two submissions were accidentally made on the same data. This has been corrected and the relevant PRIDE accession number is noted. The data will be made publicly available upon acceptance of our manuscript.

(2) Several controls are missing and data are often overinterpreted. Issues are discussed here based on the example of Fig. 4a, 1b and S1c. The issues should be solved throughout the manuscript:

a. In 4a:

- the authors claim based on 4E-BP1-T37/46 phosphorylation that rapamycin does not inhibit mTORC1 and translation, whereas the ATP analogue mTORC1 inhibitor Torin-1 does. The effect shown on 4E-BP1-pT37/46 is very well described and it is known to be mediated by the specific mode of 4E-BP1 binding to mTORC1 (PMID: 33852892). The data does not allow to conclude that rapamycin does not inhibit translation; nor does the data with Torin-1 allow to conclude that the Torin-1 effect is mTORC1 or translation mediated, as Torin-1 inhibits both mTOR complexes, and it has off target effects e.g. on PI3 kinases. The authors should correctly present literature knowledge and avoid overinterpretation of their data.

Indeed, the effect of Torin-1 in this context could also inhibit PI3K as the reviewer notes. But Torins also completely reverse GCN2 signaling (Brüggenthies et al. ref 67). The point of this figure panel is merely to show the effects of both drugs in this setting, as the effects of Torin-1 are important in the subsequent panels/experiments.

- Furthermore, quantification and statistics as well as indication of the number of replicates is missing and need to be added.

Replicate numbers and statistics were extensively and carefully recorded in the figure legends.

- Also, the total level is missing for 4E-BP1, and total levels should be detected throughout for all phosphorylated proteins.

- The loading control is missing.

- The significance of TSC2 detection in this figure is unclear as it is not adequately discussed. Either it should be discussed or left out.

TSC2 was used as the loading control in this experiment because it was included as part of the original setup, and its expression is not affected under these conditions. The point of this small figure was to show the effect of Torin-1 versus rapamycin on puromycin incorporation and phosphorylated 4EBP1.

b. Fig. 1b: The puromycin assay shows that protein synthesis is reduced upon Leu or Arg starvation, yet there is a difference between Arg and Leu starvation that is not discussed. Please provide quantification, statistics and number of replicates. Furthermore, Grb2 obviously is not an appropriate loading control as it is downregulated upon AA starvation. Other proteins, such as Tubulin, should be tested for their suitability as loading controls.

The reviewer asks multiple questions here. First, there is no true loading control that can be used in an amino acid starvation experiment – as we demonstrate here, massive changes occur to the proteome. One cannot accurately choose a loading control that is truly unchanged (by effects of translation, proteasome activity, lysosome activity and autophagy). Grb2 is our standard internal control but we know it can change, as can actin, tubulin, etc. As far as the differences in puromycin incorporation, the reviewers ask for speculation that cannot be answered with precision. Leucine may be recovered by proteasome/autophagy in greater amounts than arginine, thus driving slight increases in the relative translation rate. Further, the response to arginine and leucine starvation differs in translation control via codon usage, as noted in Darnell et al. (Ref. 41), and via other differences in TSC complex activity as shown in Carroll et al. (Ref. 29).

c. Fig. S1c: GCN2 and ATF4 are known to act in the integrated stress response. Accordingly, GCN2 is phosphorylated and ATF4 levels are enhanced after 3 h of the different types of AA starvation. However, ATF4 levels reach the peak after 3 h of starvation and then decline and in the case of Leu starvation, ATF4 vanishes after 24 h. The divergence should be discussed and detections of phospho-GCN2 should be shown for longer time points.

We are unclear on how this would aid in the interpretation of our experiments. This manuscript is not about GCN2 or ATF4. The ISR is a factor in the overall response to amino acid starvation but elaboration of the precise effects of GCN2 activation, ATF4 activation, stability and activity is an

extraordinarily complex experimental undertaking. With respect, we feel that expanding on this point, raised in the context of a supplementary figure, would go beyond the scope of the current study and would not significantly enhance the overall message of the manuscript.

(3) Legends

- Fig. S3c. What does the legend refer to? There are 3 graphs but only one legend. Please revise.

The three graphs in Fig. S3c correspond to the three time points (0, 3, and 10 hours), as indicated in the figure. It is possible that this was not immediately clear, and we revised the legend to eliminate any potential confusion.

- Fig. 3b. The figure seems to refer to 3a, yet the legends don't match. Please revise.

The legends to 3a and 3b are accurate.

(4) Terminology: Line 7: "protein translation" - proteins are not being translated. The correct term is either mRNA translation or protein biosynthesis.

Line 7 is empty (it is on the cover page). Nevertheless, we have scanned the manuscript for the term and changed accordingly.

Reviewer #2

Major comments

- Fig. 1a-c: The authors observe no strong differences in the response of cells to Leu or Arg deprivation. Given that Leu is an essential amino acid, while Arg is not (ie it can be synthesized by cells through metabolic adaptations), how do the authors explain these effects? What happens to pathways involved in arginine metabolism in Arg-deprived cells? The authors should at least discuss this further in the text.

Arginine is effectively an essential amino acid for ES cells and in fact for most cells (e.g., classic reviews from Syd Morris PMID 27934648 and related reviews). ES cells express ASS1 and ASL at the mRNA or protein level (Rebuttal figure 2, ASS1 and ASL are detected in the proteomes but at low levels and are not regulated by starvation, see our previous paper on this topic PMID 22980328). Further, other papers note arginine is an "essential" amino acid for ES cells (Refs. 29, 37). The Reviewer's point is well taken however and therefore we looked more carefully at our metabolomic data. As shown in Rebuttal Figure 4, ES indeed have detectable citrulline and argininosuccinate, the substrates of the Ass1-Asl pathway. However, as soon as arginine is limiting, both citrulline, argininosuccinate and fumarate amounts drop and do not recover, including with leucine starvation. Similarly, urea amounts drop to zero – as the urea can only come from the mitochondrial urea cycle (i.e., Arg2, not Arg1), our overall conclusion is that the any ability of the starved ES cells to regenerate arginine via Ass1-Asl or increase arginine catabolism via the urea cycle (if it exists) is limited. This finding, in our interpretation is consistent with the "shutdown/quiescence" model of reversible amino acid starvation. However, we briefly note this point at the beginning section of the Results (pg. 5, line 3) but we don't want to include these data because the manuscript is not about the specifics of the urea cycle/arginine metabolism.

- Page 9, lines 22-26, the authors mention that Torin1 triggers translation inhibition similar to that caused by Arg or Leu starvation. Moreover, starvation of either AA is sensed via mTORC1, which is inhibited by Torin1. Yet, Arg or Leu starvation fail to prevent 'threoptosis', while CHX treatment or Torin1 treatment is capable of blocking it. Is the proteome differently rewired between these nutritional and pharmacological strategies? How do the authors explain this discrepancy?

The Reviewer is right about this conundrum i.e., the difference between Arg or Leu starvation and the effects of Torin-1. While this cannot be addressed without extensive experimentation, the point is well-taken.

- A recent study described the separation of lysosomal and cytoplasmic mTORC1 functions towards distinct substrates. In line with the findings described here, Leu/Arg seem to primarily control lysosomal mTORC1, whereas Thr/Cys signal to non-lysosomal mTORC1 complexes (PMID: 39385049). How is starvation of the different AA expected to affect phosphorylation of lysosomal and non-lysosomal mTORC1 targets in this study? Minimally, the authors should discuss how their findings fit with previous work on AA sensing.

In the Fernandes et al. paper, different approaches were used to demonstrate the multiple pools of mTORC1 complex exist, and these are in a complex and/or dynamic equilibrium with the cytoplasmic mTORC1 pool. As long-suspected, Fernandes et al. provided strong data to show that the pools of mTORC1 differentially regulate the lysosomal vs. cytoplasmic substrates and are themselves differentially regulated. Fernandes showed that Cys and Thr signal via the non-lysosomal complexes in their assay platform (1 hr). Translating this system to ours is complicated by the fact that deprivation of these AAs triggers the two death pathways. Nevertheless, for -Thr we see also that pS6K is not strongly affected as shown in Rebuttal Figure 5; later on (after 3 hr) the cells begin to die, which is reflected in the total protein amounts. As far as the other amino acids, we also note substantial differences in the "rebound" phosphorylation of mTORC1 targets depending on the specific amino acid. Simplistically, we attribute this effect to mobilization of amino acids from cellular stores, which transiently activates mTORC1. Clearly, this process is extraordinarily complex. Nevertheless, the idea that different amino acids activate different mTORC1 pools is a novel advance in this field and perhaps the basis for more comprehensive approaches. Therefore, we modified the Discussion to note the potential for differential lysosomal vs. cytoplasmic signaling and how this pathway could differentially regulate the cellular response to amino acids (pg. 13, lines 4-10).

- Fig 1a: Can the authors explain why Ki67 intensity increases between 1h and 8h in the +AA samples? Does this mean that the main difference between the two time points is the changes under AA-replete conditions and not the response to Arg or Leu removal?

In this type of experiment, the fraction of Ki67+ ES cells increases in the control conditions, which reflects the fact that ES growth in a given culture is not uniform across time, as would be expected.

In standard culture conditions, ES cell growth is not synchronized, and the proportion of actively cycling cells can vary over time. This variability is a known characteristic of ES cell populations and does not indicate a specific biological change. However, the slight increase in Ki67 under +AA conditions does not obscure the clear differences observed in the -AA groups.

- Fig 1c: Why does AA re-addition at 10 and 24 hours boost cell confluence to a higher extent compared to non-starved cells (0 h time point of re-addition)?

During the starvation period, cells accumulate in a more synchronized quiescent state, and upon refeeding, re-enter the cell cycle more uniformly, contributing to the sharper increase in confluence.

- Fig. 4: The identification of a potentially novel mode of cell death ('Threoptosis') is intriguing. However, molecular players that are involved in this process are not investigated in this study. As blockage of translation prevents threoptosis, this could include translation initiation or elongation factors, mTOR signaling components, or ribosomal proteins. Although this is likely beyond the scope of this work, what are the proteomic and/or metabolic changes that occur if one compares 0h to 8h of Thr deprivation (ie before cell death is reported)? Such an approach would likely identify factors specifically related to 'threoptosis'.

First, we have changed the name to "threonine deprivation-associated cell death". Isolating the molecular pathways and players will require a genetic screen, which, as the Reviewer notes, is beyond the scope of the manuscript. **Changes are highlighted in multiple places in the manuscript.**

- Fig. 4: Removal of Met demonstrates a protective effect in Thr-starved cells, preventing the activation of 'threoptosis'. How can the authors exclude that this effect is not due to mTORC1 inhibition by Met removal, eg via the previously described SAMTOR-related pathway?

SAMTOR expression is undetectable in ES cells (in all omics modalities). See also Rebuttal Figure 2.

- Fig 4a-b: Given the effect of Torin1 but not Rapamycin in cell proliferation, as well as its known role in cell survival, can the authors exclude the involvement of mTORC2 in 'threoptosis'? Or is this due to the difference between rapamycin and torin1 in the regulation of rapamycin-resistant substrates like 4E-BP1? Please discuss this further in the text.

We decided to address this question experimentally with a systematic series of experiments using -Thr where rapamycin, rapalink, torin-1, sapanisertib or -Met was used. The results of these experiments were consistent with the results in the original submission: -Thr rapidly causes cell death as expected and this was rescued by re-addition of Thr as a control, torin-1, rapalink or sapanisertib. Rapamycin had no effect on modulating the -Thr induced death. The results with rapalink are the most informative here, as the effects of this conjugate inhibitor are mTORC1-specific. These data (from three independent experiments) are reported in Fig. S6d. Thus, the differences between rapamycin vs. rapalink/torin-1/sapanisertib suggest with reasonable confidence that mTORC1 is the main player, and that rapamycin-sensitive targets have a less role.

- The authors fail to cite and discuss a relevant recent study that described the effects of Leu and Arg availability in mESC proliferation, cell cycle arrest and pluripotency (PMID: 36430764). How do these data fit with the work presented here?

We now cited this paper (which appeared at the same time as the first author left the laboratory, **now ref. 36**). Correia et al. is similar to Todorova et al. (Ref. 37) in that the authors focus on proliferation ES cell pluripotency. Collectively, the Correia et al. and Todorova et al. papers are consistent with the cell cycle/proliferation aspects of our study.

Minor comments

- Fig S3c: The annotation of ERK1/2 and mTOR/S6K substrates is missing from the plots (all data points are gray).

This was an error on our part as a previous version of the manuscript covered some aspects of specific substrates; the annotation note in S3c is now removed.

- Fig 1f: It would make the figure more reader-friendly if the different data point groups were also labelled directly in the plots.

We have modified the figure to reflect this request. Changes in Fig. 1f.

- Fig S5d: The x-axis in the lower Leu-Cys plot is mislocalized at 100 instead of 0.

This was an error on our part, which is now corrected.

- Fig S6c: The figure legend does not seem to correspond to the data shown in the panel.

This was an error on our part and has been corrected.

- Fig 4g is wrongly referred to as 4h in the text.

This was an error on our part and has been corrected.

Reviewer #3

- The more detailed analysis of the biological consequences and mechanisms behind the responses to different amino acids are beyond the scope of this work. Still, the authors show very interesting results with the dual Threonine/Methionine removal. I wonder if they would observe a similar death (with Thr) and rescue (with Met) in cells KO for THD. Investigating the role for this enzyme in cell death triggered by Thr may be relevant for human cells - that as they mention - do not express this enzyme.

This is a good point but cannot be answered with a straightforward loss-of-function experiment as Tdh is an essential gene in ES cells (and mice).

- I am not a big fan of finding a new name for cell death processes. But that is more a personal preference. While the authors present evidence to rule out mechanisms such as apoptosis and necroptosis, there are already many other forms of cell death. Is it possible that cells fail to sense the lack of Thr and thus they continue translation as normal which then causes lethal ER stress? That would be consistent with the effects of Torin and the reduction of translation.

As noted above, we have refrained from using “threoptosis” in the revised version. As noted for the response to Reviewer 2, we are currently pursuing approaches to identify the pathways and molecular players involved in threonine-deprivation death.

- It would be nice have some additional cross-talk between their omics and more mechanistic studies. For example, do they detect an increase in GCN2 phosphorylation during -Arg and -Leu in their phosphoproteomic analysis?

pGCN2 T899 is not detected in the phosphoproteomes, which may be related to the abundance of the peptides or their detection with the present technology.

- A small detail but it may be better to mention that Arg is a conditional essential amino acid? Also, it may be worth mentioning that ES cells have special amino acid needs/mechanisms including the expression of glutamine synthase and thus are able to survive without Gln supplementation.

See the response to the first comment of Reviewer 2. We have slightly modified the relevant sections to reflect the specific requirements of ES cells for specific amino acids.

- Are columns in Fig. 1h/I also ordered by hierarchical clustering? It is fine either way but to me it is not clear if the column order emerges from the data, or the columns are manually organized by treatment/temporal response. Again, I think either would be OK.

The columns are organized by treatment and time and the rows by hierarchical clustering. As this may have been unclear, we modified the legend to be clearer. Changes pg. 36.

- In Figure 1g, it is unclear what the gray boxes represent (maybe the number of proteins/phosphoproteins/transcripts that were not significantly different?). I'd suggest clarifying this. In the same figure, in the left plot, there is a lost number (2041) below the blue box in the middle. I'm suspecting it belongs in the second panel, on top of the blue box in the middle.

We corrected this figure as noted by the reviewer. The grey boxes represent the number of common changes (up or down between -Arg vs.-Leu). We modified the figure legend to be clear on this point.

- In the GCN2 blots shown in S1 it would be better to use total GCN2 to highlight the increase in phosphor GCN2.

This experiment would be ideal (e.g., to show a shift with phosphorylation while controlling the amount of total GCN2) but it is not possible with the current reagents and gel systems. First, GCN2 is activated by the T899-p event (i.e., a single event) in a 189 kDa protein. Such a difference cannot be resolved (see Brüggenthies et al. Ref. 66 for examples in a different cell system). Second, the total anti-GCN2 antibody does not perform well in murine ES cells, including by phosho-flow using permeabilized cells. We have spent, and continue to spend, significant efforts in antibody validation for the ISR pathway. As the RNAseq data from the GCN2 KO gives the expected results, and that GCN2 is not a major focus of the manuscript, we feel that the image in S1c is sufficient for the reader to understand the activation kinetic of GCN2 in this system.

- Figure S3c: there is a legend for phospho-sites from Erk1/2 and mTOR/p70s6kb substrates, but no colored dots are visible in the plot (that I can see at least).

This was an error on our part as a previous version of the manuscript covered some aspects of specific substrates; the annotation note in S3c is now removed. Changes pg. 47.

August 22, 2025

RE: Life Science Alliance Manuscript #LSA-2025-03324R

Prof. Peter Murray
Max Planck Institute of Biochemistry
Immunoregulation Research Group
Am Klopferspitz 18
Martinsried 82152
Germany

Dear Dr. Murray,

Thank you for submitting your revised manuscript entitled "Differential cell survival outcomes in response to diverse amino acid stress". As you will see, the reviewers feel that their main concerns have been addressed in this revision. Please consider the remaining points raised by Reviewer 1, which can be addressed by amending the text as needed. We would be happy to publish your paper in Life Science Alliance pending these changes and final revisions necessary to meet our formatting guidelines.

- Please upload your main and supplementary figures as single files.
- It is recommended to exclude figures from the manuscript text and upload them separately.
- Please add the X and Bluesky handles of your host institute/organization as well as your own and/or one of the authors in our system.
- Please be sure that the authorship listing and order are correct and match between the system and the manuscript file.
- Please label the "Summary" as the Abstract.
- Please use the [10 author names, et al.] format in your references (i.e., limit the author names to the first 10).
- Please consult our manuscript preparation guidelines <https://www.life-science-alliance.org/manuscript-prep> and make sure your manuscript sections are in the correct order.
- Please add your main and supplementary figure legends to the main manuscript text after the references section.
- Please add callouts for Figure S3A-B to your main manuscript text.
- Please add molecular weight markers to blots in Figure 4A and Figure S1C.

A. FINAL FILES:

B. MANUSCRIPT ORGANIZATION AND FORMATTING:

Sincerely,

Reviewer #2 (Comments to the Authors (Required)):

I appreciate the authors' efforts to revise the manuscript. The changes generally help to cover most of my previous concerns, also taking into account that it is not expected by this particular journal that they expand their focus too much. I have few, but important, remaining requests that I believe should be sufficiently addressed prior to acceptance of this, otherwise, very interesting paper.

1) With regards to my previous comment #2 (both Arg/Leu starvation and Torin1 treatment inhibit mTORC1, yet only the latter can potently block Thr-starvation induced cell death), the authors responded that they "have modified the Discussion to reflect the issue raised". Unfortunately, however, I was not able to find a section that explicitly refers to this apparent conundrum in the text. Although I acknowledge that it would be beyond the scope of this study to experimentally address this discrepancy, to avoid confusing the reader, my strong suggestion would be that the authors expand their discussion to sufficiently and more clearly cover this part in the final version of the manuscript.

2) With regards to my previous comment #8 (differential effect of rapamycin versus Torin in cell proliferation, shown in Fig. 4a-b, and in preventing Thr-starvation-induced cell death), the authors have now provided additional data (new Fig. S6d) comparing Torin1, rapamycin, sapanisertib, and rapalink in their ability to prevent Thr-starvation-induced cell death. Based on the fact that rapalink rescues cell death, whereas rapamycin doesn't, the authors conclude this effect is via the inhibition of mTORC1 and likely due to its action on rapamycin-resistant targets. Although, as stated in my previous comment #8, I fully agree that 4EBP1 is a plausible candidate, because it is a rapamycin-resistant mTORC1 substrate and because of its well-described role in the regulation of protein synthesis, I do not agree that the data are sufficient to exclude mTORC2 as a potential player. This is because, rapalink can also downregulate the phosphorylation of 4EBP1 (both rapamycin-sensitive and -resistant sites) and of mTORC2 targets like AKT when used at somewhat higher concentrations and/or for longer times. Therefore, its use does not by definition exclude an effect on mTORC2 as well, especially given that the authors do not provide any information about these treatments, or control immunoblots that would show an mTORC1-specific effect.

For this claim to be made, it would be absolutely necessary to: 1) add in the methods and in the figure legends detailed information about drug concentrations and duration of treatments, and 2) perform immunoblotting experiments with samples

treated as in S6d showing S6K, 4EBP1 and AKT(S473) phosphorylation (including total protein controls) as read-outs for mTORC1/2. If rapalink also affects mTORC2 activity under the conditions tested, the data shown in S6d would not really address the issue raised previously.

Alternatively, the authors could simply acknowledge in the discussion that although the currently available data suggest a role for 4EBP and presumably other rapamycin-resistant mTORC1 targets in this phenomenon, they leave open the possibility that mTORC2 could also be involved, especially given its well-known role in controlling cell death-related pathways.

Reviewer #3 (Comments to the Authors (Required)):

I reiterate my consideration that this is a valuable piece of work. The authors also strived to address my prior concerns so I have no further suggestions.

August 25, 2025

RE: Life Science Alliance Manuscript #LSA-2025-03324RR

Prof. Peter Murray
Max Planck Institute of Biochemistry
Immunoregulation Research Group
Am Klopferspitz 18
Martinsried 82152
Germany

Dear Dr. Murray,

Thank you for submitting your Research Article entitled "Differential cell survival outcomes in response to diverse amino acid stress". It is a pleasure to let you know that your manuscript is now accepted for publication in Life Science Alliance. Congratulations on this interesting work.

DISTRIBUTION OF MATERIALS:

Again, congratulations on a very nice paper. I hope you found the review process to be constructive and are pleased with how the manuscript was handled editorially. We look forward to future exciting submissions from your lab.

Sincerely,
